# Assessment of Regional Aerosol Radiative Effects under SWAAMI Campaign – PART 2: Clear-sky Direct Shortwave Radiative Forcing using Multi-year Assimilated Data over the Indian Subcontinent

Harshavardhana Sunil Pathak[1], Sreedharan Krishnakumari Satheesh[1,2], Krishnaswamy Krishna Moorthy[1], and Ravi Shankar Nanjundiah[1,2,3]

[1]Centre for Atmospheric and Oceanic Sciences, Indian Institute of Science, Bangalore, India
[2]DST-Centre of Excellence in Climate Change, Divecha Centre for Climate Change, Indian Institute of Science, Bangalore, Bengaluru 560012, India
[3]Indian Institute of Tropical Meteorology, Pune 411008

**Correspondence:** Harshavardhana Sunil Pathak (rhsp19@gmail.com)

**Abstract.** Clear-sky, direct shortwave Aerosol Radiative Forcing (ARF) has been estimated over the Indian region, for the first time employing multi-year (2009-2013) gridded, assimilated aerosol products, as an important part of the South West Asian Aerosol Monsoon Interactions (SWAAMI) which is a joint Indo-UK research field campaign focused at understanding the variabilities in atmospheric aerosols and their interactions with the Indian summer monsoon. The aerosol datasets have

been constructed following statistical assimilation of concurrent data from a dense network of ground-based observatories, and multi-satellite products, as described in Part-1 of this two-part paper. The ARF, thus estimated, are assessed for their superiority or otherwise over other ARF estimates based on satellite-retrieved aerosol products, over the Indian region, by comparing the radiative fluxes (upward) at Top of Atmosphere (TOA) estimated using assimilated and satellite products with spatio-temporally matched radiative flux values provided by CERES (Clouds and Earth's Radiant Energy System) Single Scan

Footprint (SSF) product. This clearly demonstrated improved accuracy of the forcing estimates using the assimilated vis-a-vis satellite-based aerosol datasets; at regional, sub-regional and seasonal scales. The regional distribution of diurnally averaged ARF estimates has revealed (a) significant differences from similar estimates made using currently available satellite data, not only in terms of magnitude but also sign of TOA forcing; (b) largest magnitudes of surface cooling and atmospheric warming over IGP and arid regions from north-western India; and (c) negative TOA forcing over most parts of the Indian region, except

for three sub-regions - the Indo-Gangetic plains (IGP), north-western India and eastern parts of peninsular India where the TOA forcing changes to positive during pre-monsoon season. Aerosol induced atmospheric warming rates, estimated using the assimilated data, demonstrate substantial spatial heterogeneities ($\sim 0.2$ to $2.0$ K day$^{-1}$) over the study domain with the IGP demonstrating relatively stronger atmospheric heating rates ($\sim 0.6$ to $2.0$ K day$^{-1}$). There exists a strong seasonality as well; with atmospheric warming being highest during pre-monsoon and lowest during winter seasons. It is to be noted that

the present ARF estimates demonstrate substantially smaller uncertainties than their satellite counterparts, which is a natural consequence of reduced uncertainties in assimilated vis-a-vis satellite aerosol properties. The results demonstrate the potential

application of the assimilated datasets and ARF estimates for improving accuracies of climate impact assessments at regional and sub-regional scales.

## 1   Introduction

The uncertainties in aerosol radiative forcing (ARF) pose primary challenges in the assessment of climatic implications of atmospheric aerosols at global, regional and even sub-regional scales (Schwartz, 2004; Boucher et al., 2013). In order to improve the estimates of aerosol climate sensitivity, at least three-fold reduction in the uncertainties in aerosol radiative forcing is necessary (Schwartz, 2004). Despite efforts towards this, significant uncertainties still persist in the estimates of even direct aerosol radiative forcing (DARF) (Penner et al., 1994; Boucher and Anderson, 1995; Ramanathan and Carmichael, 2008), leave alone the indirect forcing. This calls for improvement in the accuracy of primary aerosol inputs to DARF estimation at sub-regional and regional scales.

There have been several estimates of global aerosol radiative forcing (ARF) by employing general circulation models (GCM) or chemistry transport models (CTM) making use of aerosol emission inventories (Jacobson, 2001; Takemura et al., 2002; Myhre et al., 2007, 2009; Kim et al., 2008). These studies have highlighted the regional and temporal heterogeneity in aerosol forcing, but the actual forcing values reported have significant uncertainties emanating mainly from those in the input inventories, meteorology and assumptions made in aerosol-chemistry processing. Schulz et al. (2006) have shown that the model (GCM or CTM) based estimates of radiative forcing differ significantly amongst themselves, even in terms of sign of radiative forcing despite using identical emission inventories. Recognizing this issue, Chung et al. (2005, 2010) have produced global and regional maps of aerosol forcing employing observationally constrained aerosol datasets, which were constructed by integrating satellite and ground based observations of aerosol optical depth (AOD) with those derived from the global chemistry-transport model, GOCART. They provided somewhat more realistic estimates of aerosol forcing as compared those incorporating model-derived aerosol parameters, because of the improvements in the input datasets arising from their assimilation efforts. Nevertheless, due to limited number of regional ground stations involved in the assimilation process by Chung et al. (2005), (for example only 2 stations over Indian region), the assimilated AODs over large parts of the globe remained largely represented by satellite-retrieved and model-derived AODs, which suffer from significant uncertainties and biases emanating from a variety of sources (cloud contamination, spatial heterogeneities in surface albedo, sparse temporal sampling, various assumption made during retrievals procedure and sensor degradation etc. (Zhang and Reid, 2006; Jethva et al., 2014). As a result, large uncertainties still prevailed at regional and sub-regional scales.

Location-specific estimates are more accurate as these employ highly accurate ground-based measurements of aerosol properties (spectral AOD/ aerosol absorption/ altitude profiles etc.) and generate aerosol models constrained with measurements and use them in a radiative transfer scheme (Babu and Moorthy, 2002; Babu et al., 2007; Satheesh et al., 2006; Pathak et al., 2010; Sinha et al., 2013). Due to smaller uncertainties in the direct measurements, these aerosol forcing estimates tend to have lesser uncertainties vis-a-vis forcing estimates from satellite-retrieved and model-derived aerosol parameters. However, due to limited spatial representativeness of each ground station, and the limited density of ground networks, these ARF estimates are

highly location specific. They lack the much-needed regional representativeness for climate assessment, unless they involve a large number of ground locations, from a highly dense network, which has associated practical difficulties. Recognizing these scenarios, especially over the Indian region which has large spatio-temproal variations in aerosol properties, a careful assimilation of moderately dense network data with satellite data to generate a gridded data set which is more or less continuous in space and time, has been envisioned as one of the key objectives by the South West Asian Aerosols Monsoon Interactions (SWAAMI) (Morgan, 2016) (https://gtr.ukri.org/projects?ref=NE%2FL013886%2F1), a co-ordinated field campaign jointly undertaken by the scientists from India and the United Kingdom.

In part-1 of this two-part paper, we have presented, the gridded, assimilated datasets of AOD and SSA (Single Scattering Albedo) over India, which provide spatio-temporally continuous data, generated for the first time by harmonizing long-term (2009-2013) measurements from a dense network of ground-based aerosol observatories and multi-satellite datasets following statistical assimilation techniques (Pathak et al., 2019). The resulting improvement in accuracies of the gridded products (over the parent data) in reproducing the spatio-temporal characteristics of aerosol properties at sub-regional scales over the Indian domain was also demonstrated. In this part-2 of the paper, we have estimated direct shortwave aerosol radiative forcing (ARF) over the Indian region using the above gridded data and examined its features. A comparison of these estimates is made with similar estimates made using the parent satellite data to demonstrate the effectiveness of the assimilated data in better-quantifying ARF over the Indian region with its characteristic spatio-temporal features. Further, we have compared the top-of-the-atmosphere (TOA) flux estimated using the assimilated data with the instantaneous flux values measured by Clouds and Earth's Radiant Energy System (CERES) instrument and the seasonal contrast in ARF is then presented for various geographically homogeneous sub-regions. The primary findings of the present work are then summarized.

## 2 Database

Accuracy of estimation of direct aerosol radiative effect depends strongly on the accuracies of three key optical properties of aerosols, namely AOD, SSA and single-scatter phase function, and the availability of these continuously in space and time over the domain of interest. Accordingly, in this work, we have used the gridded data over the Indian domain for AOD and SSA at $1° \times 1°$ resolution. These datasets are generated by assimilating long-term data from ground-based network observatories and space-borne sensors following statistical assimilation techniques (3D-VAR and Weighted Interpolation methods) as described in Pathak et al. (2019) (the part-1 paper) and are available at *http://dccc.iisc.ac.in/aerosoldata/*. These assimilated AOD and SSA are henceforth respectively denoted as AS AOD and AS SSA. However, such a comprehensive dataset for aerosol phase function is not available over the study domain, and as such we relied on 'Optical Properties of Aerosols and Clouds' (OPAC) model (Hess et al., 1998), which provides optical properties for variety aerosol species and their mixtures (under externally mixed assumption). As our study domain mainly comprises of land areas, an average value of aerosol phase function (at 550 nm) corresponding to all continental aerosol mixtures in OPAC is considered, and its Legendre polynomial coefficients (8 streams) are employed for determining the Legendre moments of the phase function.

In addition to assimilated aerosol products, we have also employed satellite-retrieved AOD and SSA (denoted as SR AOD and SR SSA respectively) for ARF estimation. The satellite-retrieved AODs employed here are constructed by combining by monthly averaged AOD products ($1° \times 1°$ resolution) from (L3, collection 6, *https://modis.gsfc.nasa.gov/data/*) MODerate Imaging Spectroradiometer (MODIS) onboard AQUA and TERRA satellites as well as from Multiangle Imaging SpectroRa-

diometer (MISR) onboard TERRA satellite (L3, *https://misr.jpl.nasa.gov/*) (Diner et al., 1998), as detailed in the Part-1 of the two-part paper (Pathak et al., 2019). The satellite-retrieved SSA are provided by monthly mean SSA (at 500 nm) datasets constructed from upwelling radiance measurements (in the range of 270-500 nm) performed by OMI on-board AURA satellite (Torres et al., 2002; Levelt et al., 2006).

   The spatial distribution of the gridded aerosol properties (assimilated as well as satellite-retrieved; AODs and SSA) are

shown in Figure 1 for two typical months; January - typical winter, when majority aerosols are trapped near the surface due to the shallow atmospheric boundary layer; and May - typical summer/pre-monsoon when the strong thermal convection ensures a thorough vertical mixing within the deep Atmospheric Boundary Layer (ABL) (Kompalli et al., 2014). These two months also signify periods when the local emissions dominate in the aerosol loading (winter) and when advected aerosols (dust and sea-salt) also contributes significantly to the aerosol loading (summer/pre-monsoon) (Moorthy et al., 2005, 2007b; Jethva et al.,

2005; Niranjan et al., 2007). Sub-regional differences in the spatial distribution of these parameters between the assimilated and satellite retrieved are noticeable in Figure 1; the most conspicuous being in SSA where the assimilated data shows stronger absorption than the satellite retrieved over many parts of the domain.

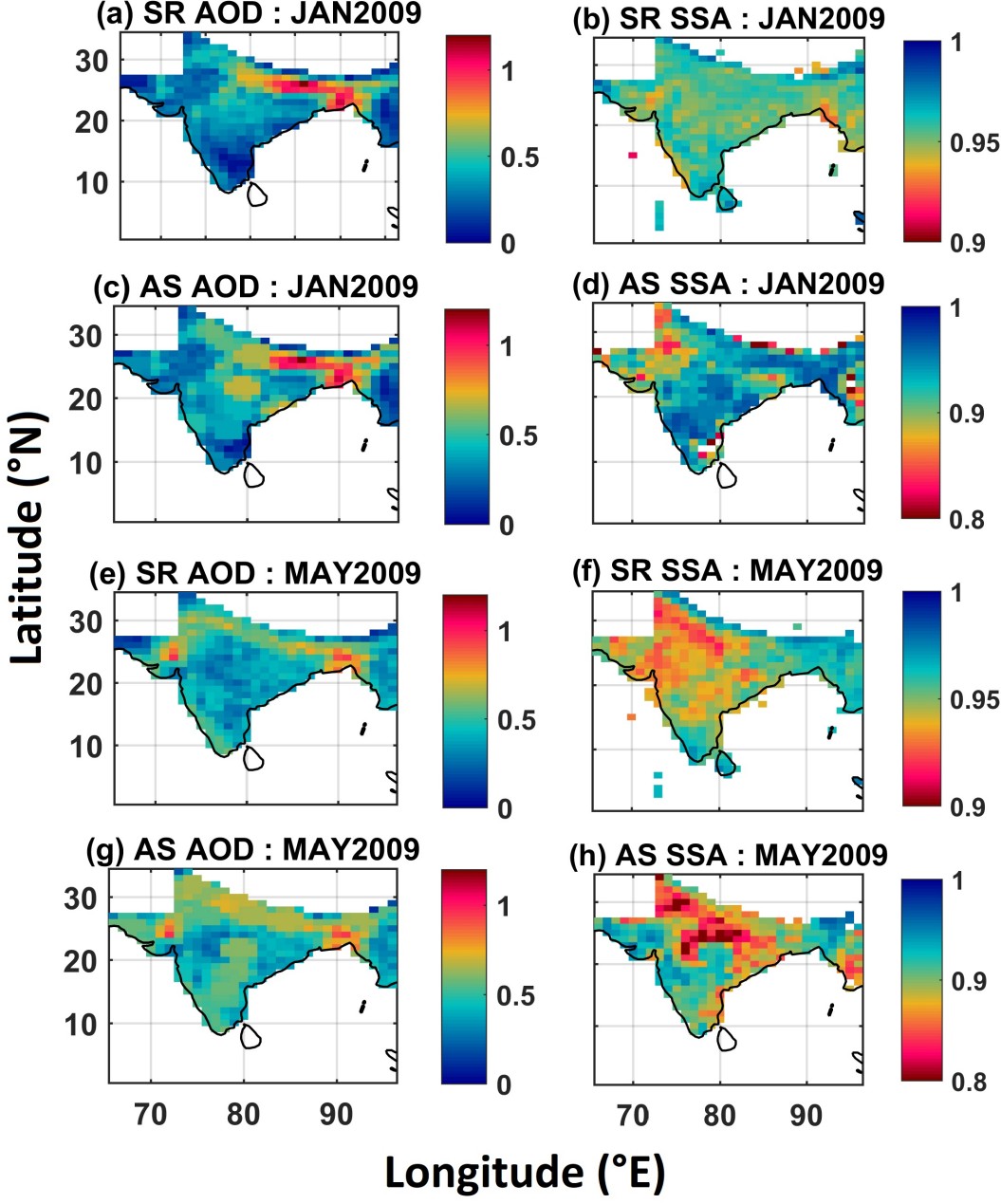

**Figure 1.** Spatial Variation of (a) SR AOD for Jan-2009, (b) SR SSA for Jan-2009, (c) AS AOD for Jan-2009, (d) AS SSA for Jan-2009, (e) SR AOD for May-2009, (f) SR SSA for May-2009, (g) AS AOD for May-2009 and (h) AS SSA for May-2009, over the Indian region

## 3 ARF estimation

For estimating aerosol radiative forcing, we have used the Santa Barbara DISORT Atmospheric Radiative Transfer (SBDART) radiative heat transfer model, which uses the DIScrete Ordinate Radiative Transfer (DISORT) algorithm to solve the radiative heat transfer equation with plane-parallel assumption, in the atmosphere with vertical inhomogeneities (Ricchiazzi et al., 1998). The above described columnar AOD and SSA (section 2) data are used as inputs. Here, the aerosols are considered to be exponentially distributed in vertical direction with the typical scale height of 1.45 km (Ricchiazzi et al., 1998). The surface reflectance data needed has been taken from MODIS, while vertical distribution of atmospheric gases, except columnar Ozone and water vapour, is specified using the tropical environment model provided by SBDART. For columnar ozone and water vapour, datasets for the corresponding period provided respectively by OMI and MODIS are used (Ziemke et al., 2006; Gao and Kaufman, 2003). The upward and downward shortwave fluxes at the TOA and surface, (in the wavelength range 0.2 to 4 $\mu$m) are computed using SBDART for each hour from 6 am (approximate local sunrise time in IST) to 6 pm (approximate local sunset time in IST) for each grid point. The net radiative fluxes are then estimated (considering upward negative and downward positive) for 'with aerosol' and 'without aerosol' conditions and then ARF is estimated as the difference between the net fluxes for the two conditions as has been described in Equations 1-2.

$$\text{AS ARF}^*_{\text{TOA}} = (F\downarrow_{\text{TOA}} - F\uparrow_{\text{TOA}})_{\text{with aerosol}} - (F\downarrow_{\text{TOA}} - F\uparrow_{\text{TOA}})_{\text{no aerosol}} \tag{1}$$

$$\text{AS ARF}^*_{\text{srf}} = (F\downarrow_{\text{srf}} - F\uparrow_{\text{srf}})_{\text{with aerosol}} - (F\downarrow_{\text{srf}} - F\uparrow_{\text{srf}})_{\text{no aerosol}} \tag{2}$$

Here, AS ARF$^*$ denotes aerosol radiative forcing estimated using assimilated aerosol properties while the subscripts TOA and srf denote the top of atmosphere and ground surface respectively for a given grid point and solid angle. The upward and downward fluxes over the shortwave spectrum are respectively denoted by $F\uparrow$ and $F\downarrow$. The corresponding atmospheric forcing (denoted by AS ARF$^*_{\text{atm}}$) due to aerosol absorption is then estimated as shown in the following equation 3.

$$\text{AS ARF}^*_{\text{atm}} = \text{AS ARF}^*_{\text{TOA}} - \text{AS ARF}^*_{\text{srf}} \tag{3}$$

The ARF values estimated in Equations 1-3, which are specific to a solar-zenith angle (or time of the day) for a given location, are further averaged (over the period of 12 hours) and then halved in order to estimate the diurnally averaged shortwave ARF for the given grid point. These diurnally averaged ARF at TOA, surface and in atmosphere are henceforth referred as AS ARF$_{\text{TOA}}$, AS ARF$_{\text{srf}}$ and AS ARF$_{\text{atm}}$ respectively (without asterisk).

With a view to assessing the improvements (or otherwise) of the current estimates over such estimates made using conventional datasets (satellite retrieved or ground measured), we have also estimated direct ARF by using only the satellite-retrieved aerosol products (AOD and SSA) in SBDART for the same period, over the same domain. All other input parameters are kept

the same as for AS dataset. The diurnally averaged ARF estimates corresponding to satellite products, at TOA, surface and within atmosphere are henceforth referred to as SR $ARF_{TOA}$, SR $ARF_{srf}$ and SR $ARF_{atm}$ respectively.

Further, from each of the forcing estimates made using the two different datasets, we also estimated the difference between the two (referred to as dARF) as,

$$dARF_{TOA} = AS\ ARF_{TOA} - SR\ ARF_{TOA} \tag{4}$$

$$dARF_{srf} = AS\ ARF_{srf} - SR\ ARF_{srf} \tag{5}$$

$$dARF_{atm} = AS\ ARF_{atm} - SR\ ARF_{atm} \tag{6}$$

## 4 Results and discussion

### 4.1 Spatial distribution of aerosol radiative forcing

The regional distribution of the different components of aerosol radiative forcing, estimated as above, are shown in figure 2 to 4 for the representative winter month (January) and in figures 5 to 7 for the representative pre-monsoon month (May). Each figure shows the spatial distribution of ARF determined using assimilated datasets (AS ARF), satellite retrieved datasets (SR ARF) and the difference between the two; for TOA, surface and atmosphere. As the dARF calculation has to account for the signs of the respective ARF estimates, positive $dARF_{TOA}$ (which is estimated as shown in equation 4) indicate smaller magnitudes of AS $ARF_{TOA}$ vis-a-vis SR $ARF_{TOA}$ and vice-versa when both RF estimates are negative, which is the case in general. The same can be said for $dARF_{srf}$ (equation 5), the difference between two surface forcing estimates which are always negative. However, when both AS and SR $ARF_{TOA}$ estimates are positive, the larger (smaller) magnitudes of AS $ARF_{TOA}$ vis-a-vis SR $ARF_{TOA}$, would yield positive (negative) values of $dARF_{TOA}$.

From Figures 2-4 we can infer the following. During winter:

1. Over most of the study domain, ARF estimated using the assimilated datasets show stronger surface cooling and higher atmospheric warming than those yielded from the satellite retrieved data, except over outflow region of Indo-Gangetic plain. The higher atmospheric forcing correspoding to the assimilated dataset is mainly contributed by the lower SSA in the assimilated data (owing to assimilation with ground-based measurements of aerosol absorption).

2. Over most of the domain, the TOA forcing remains negative in both the estimates; though the magnitude is lower when assimilated data are used. However, weak positive values are seen over the arid regions of northwest, the Himalayan foothills and a small region on the eastern part of the peninsula (circled in the figure 2a).

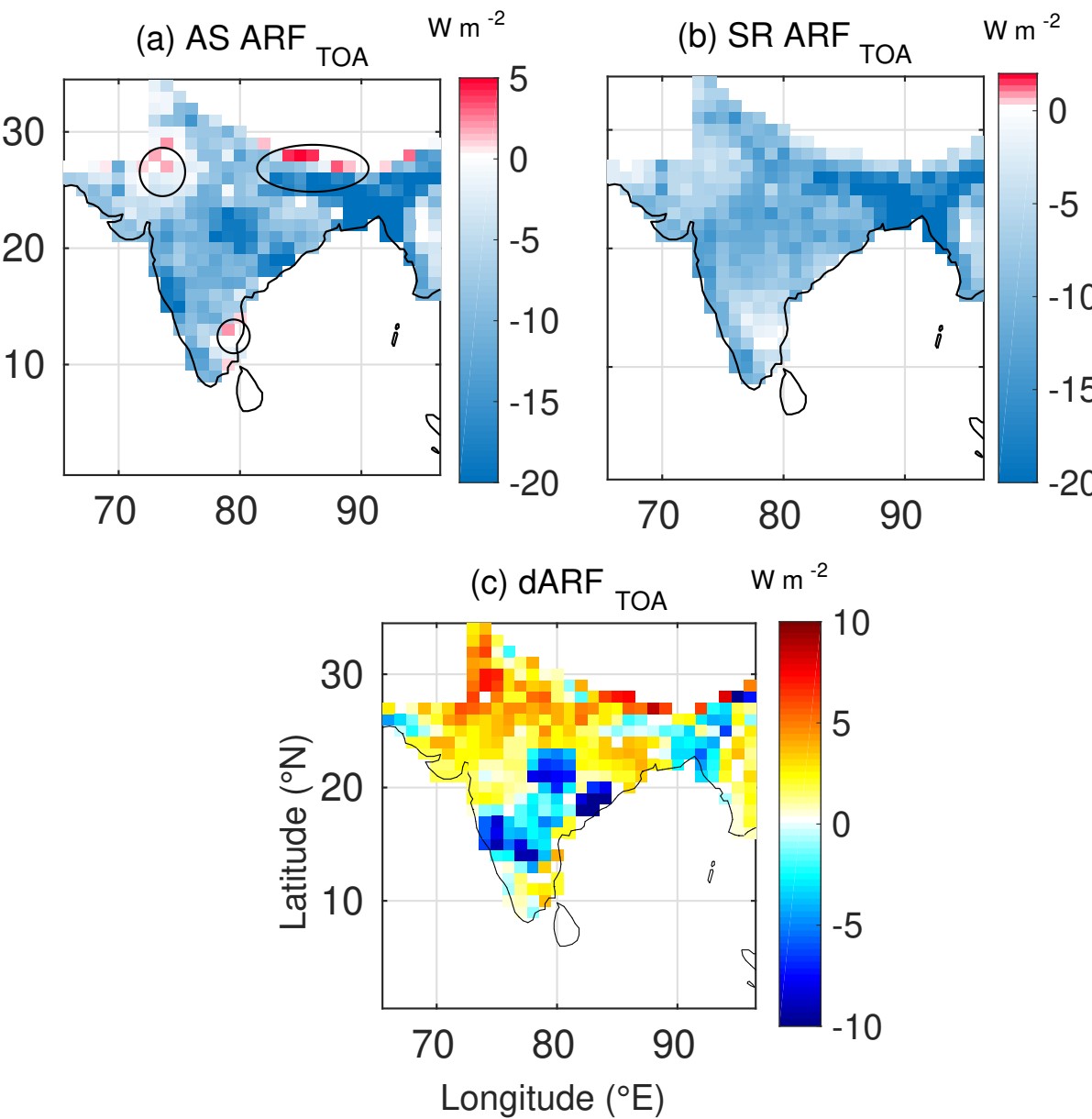

**Figure 2.** Spatial Variation of (a) AS ARF$_{TOA}$ (radiative forcing at TOA estimated using assimilated AOD and SSA), (b) SR ARF$_{TOA}$ (radiative forcing at TOA estimated using satellite-retrieved AOD and SSA), and (c) the difference between both TOA forcing estimates (dARF$_{TOA}$) for January-2009, over the Indian region

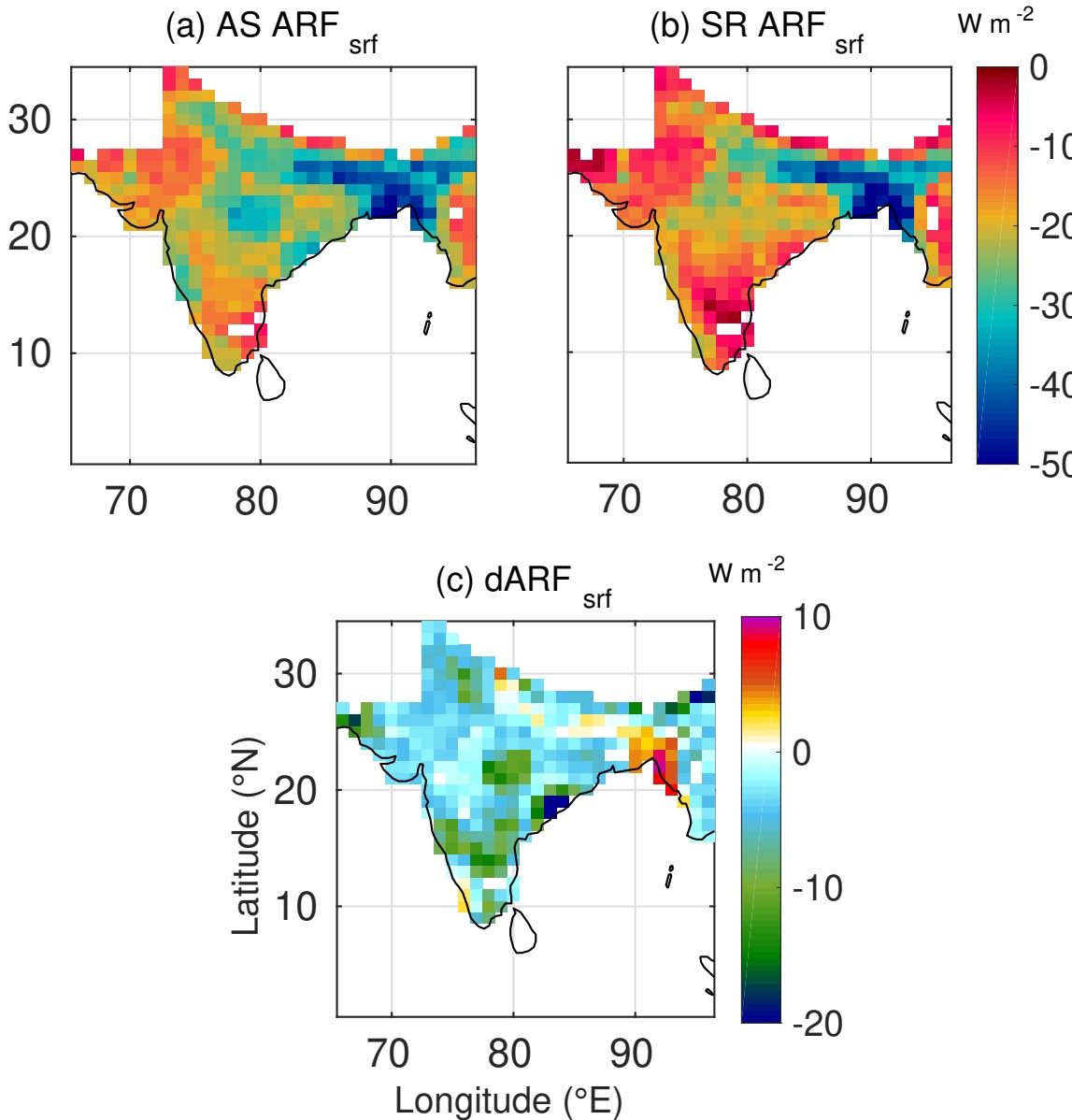

**Figure 3.** Spatial Variation of (a) AS ARF$_{srf}$ (radiative forcing at surface estimated using assimilated AOD and SSA), (b) SR ARF$_{srf}$ (radiative forcing at surface estimated using satellite-retrieved AOD and SSA), and (c) the difference between both surface forcing estimates (dARF$_{srf}$) for January-2009, over the Indian region

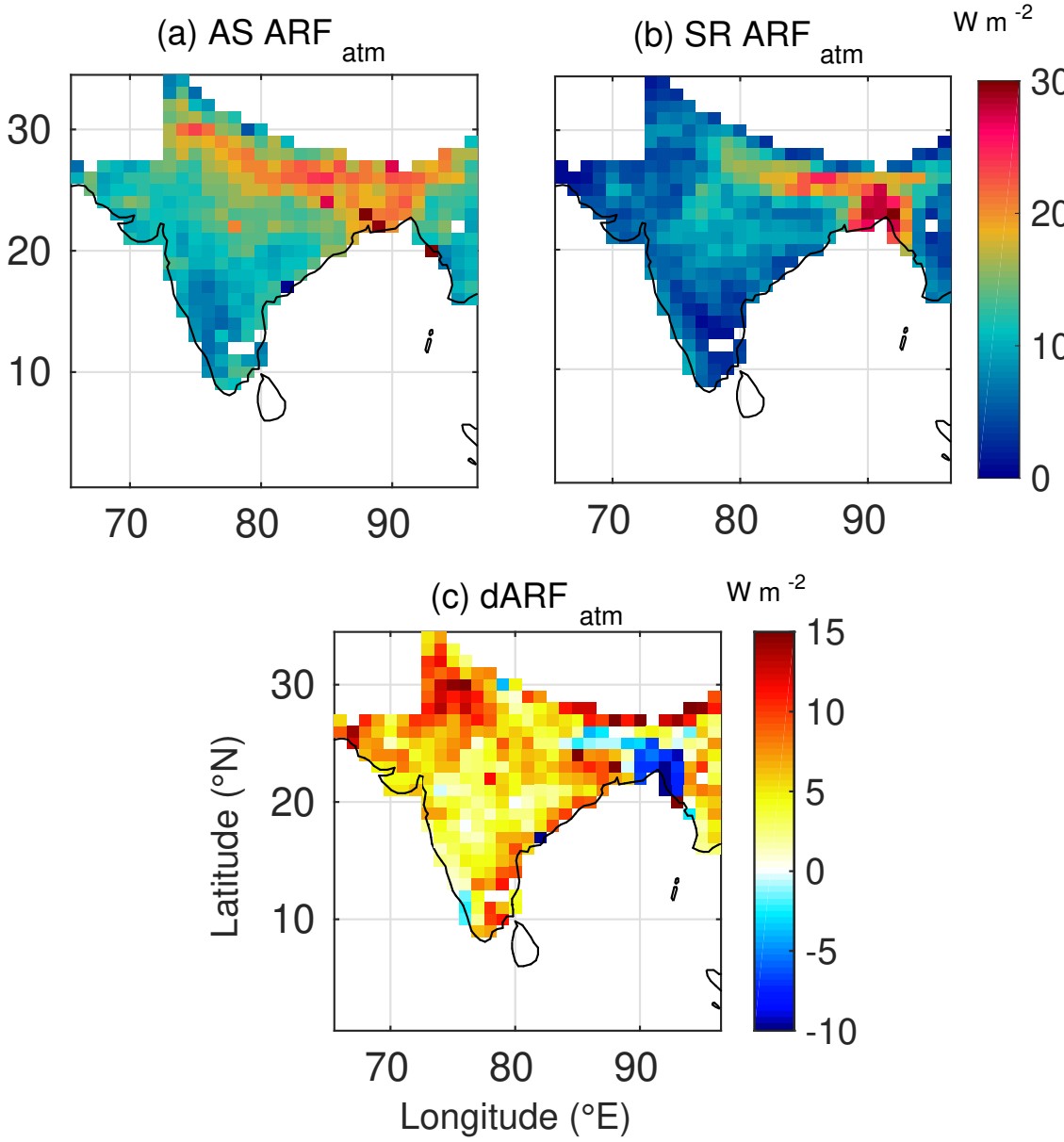

**Figure 4.** Spatial Variation of (a) AS ARF$_{atm}$ (atmospheric forcing estimated using assimilated AOD and SSA), (b) SR ARF$_{atm}$ (atmospheric forcing estimated using satellite-retrieved AOD and SSA), and (c) the difference between both atmospheric forcing estimates (dARF$_{atm}$) for January-2009, over the Indian region

During summer (Figures 5-7), the seasonal transformation of radiative impacts is clearly seen in the assimilated forcing (mainly due to seasonal change in the aerosol types with anthropogenic aerosols being abundant during winter and natural aerosols during summer). The salient features are:

1. Significantly large increase in all the three components of aerosol forcing compared to the winter-time values, primarily due to enhanced aerosol loading in summer.

2. This enhancement is seen more conspicuously in the maps using assimilated data (AOD and SSA) as inputs, than those generated using satellite derived data as input, over most parts of the region.

3. In sharp contrast to the winter case, during summer the TOA forcing using assimilated data shows positive values over a large area, with highest values over the IGP and eastern Indian landmass, primarily resulting from incorporation of more realistic values of SSA in the assimilated data. Surface forcing remains, nevertheless negative in both the estimates.

4. Consequently, higher atmospheric forcing results over the entire landmass, except for a small region in the northwest, when assimilated data is used (Figure 7c).

5. Positive TOA forcing and strong atmospheric absorption over IGP results due to the presence of absorbing aerosol species, (mainly carbonaceous aerosols and mineral dust, (Moorthy et al., 2005; Niranjan et al., 2007; Vaishya et al., 2018)), which brings down the SSA and the respective ground-based measurements reflecting the same are assimilated in the AS data sets (Pathak et al., 2019). The strong and vigorous vertical motions within the convective boundary layers, (when the surface temperatures are above 40 C during large parts of daytime) lift these aerosols to upper levels of atmosphere (Prijith et al., 2016) thereby further increasing the absorption (Chand et al., 2009). Thus, higher aerosol loading from local as well as remote sources and stronger vertical mixing within deeper planetary boundary layer (especially during pre-monsoon) over IGP could be responsible for higher ARF over Indo-Gangetic region vis-a-vis other parts.

However, it is to be noted here that the actual verification these features regarding vertical distribution of aerosols and their corresponding effects on ARF at regional level demands vertically resolved, gridded aerosol datasets instead of column-integrated aerosol products. Currently, we are in the process of assimilating the entire data from airborne measurements over the Indian region (Babu et al., 2016; Vaishya et al., 2018), carried out during different campaigns since 2007. The resulting vertically resolved and observationally constrained gridded aerosol properties would be suitable for the regional level assessment of vertically resolved ARF.

6. The positive TOA forcing values occurring over eastern Peninsular India (figure 2a and 5a) are primarily due to lower columnar SSA values (0.7 to 0.85) during winter and pre-monsoonal months as demonstrated in figure 1d and 1h respectively. These low SSA values indicate the increased presence of Black Carbon (BC) which can be largely associated with large anthropogenic activities (this region has several major harbors, industries and large urban conglomerates such as Chennai (13.08 N, 80.27 E), Vijayawada (16.51 N, 80.65 E), Visakhapatnam (17.68 N, 83.21 E), Bengaluru (12.97 N, 77.59 E), Bhubaneswar (20.30 N, 85.42 E)).

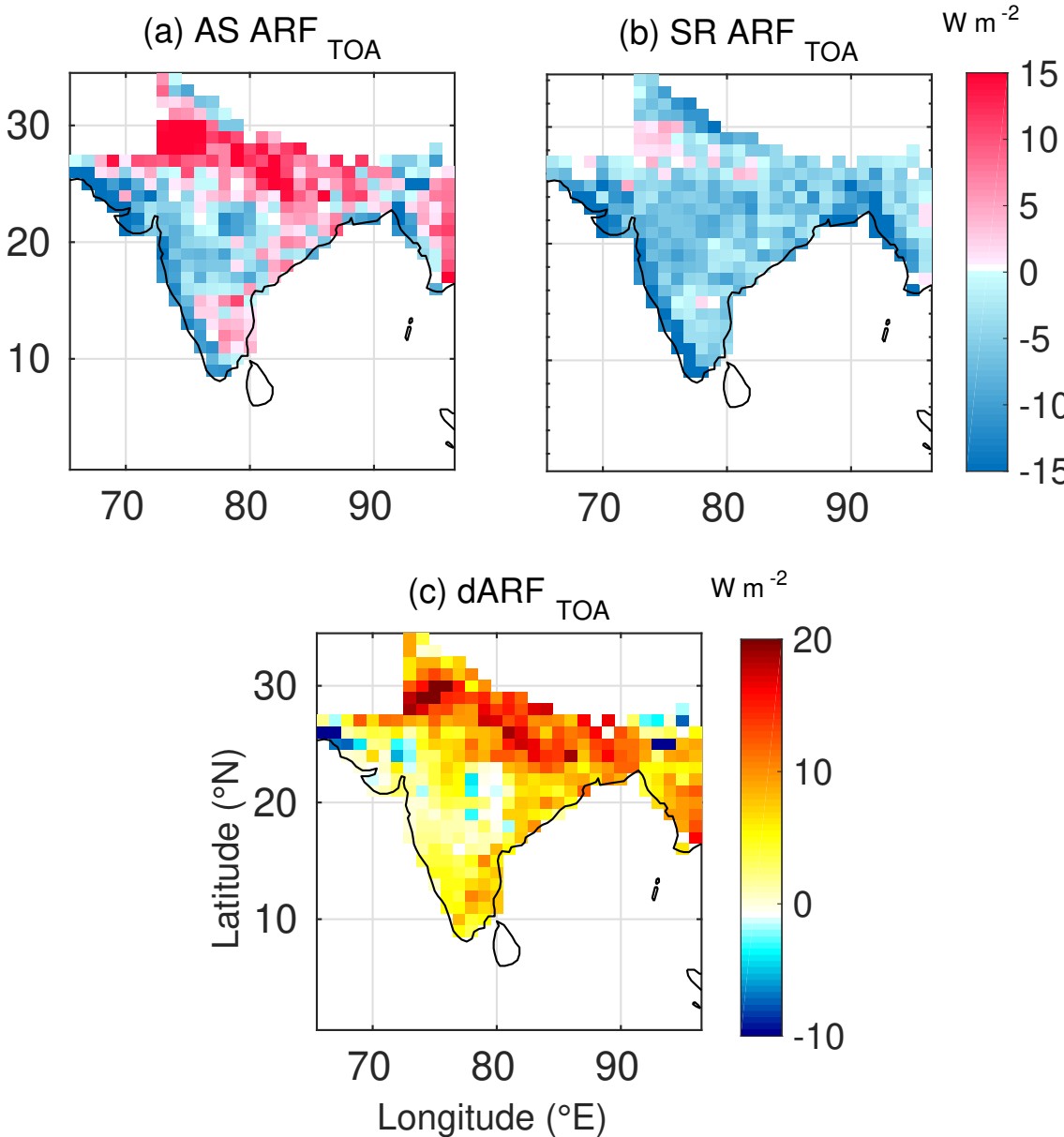

**Figure 5.** Spatial Variation of (a) AS ARF$_{TOA}$ (radiative forcing at TOA estimated using assimilated AOD and SSA), (b) SR ARF$_{TOA}$ (radiative forcing at TOA estimated using satellite-retrieved AOD and SSA), and (c) the difference between both TOA forcing estimates (dARF$_{TOA}$) for May-2009, over the Indian region

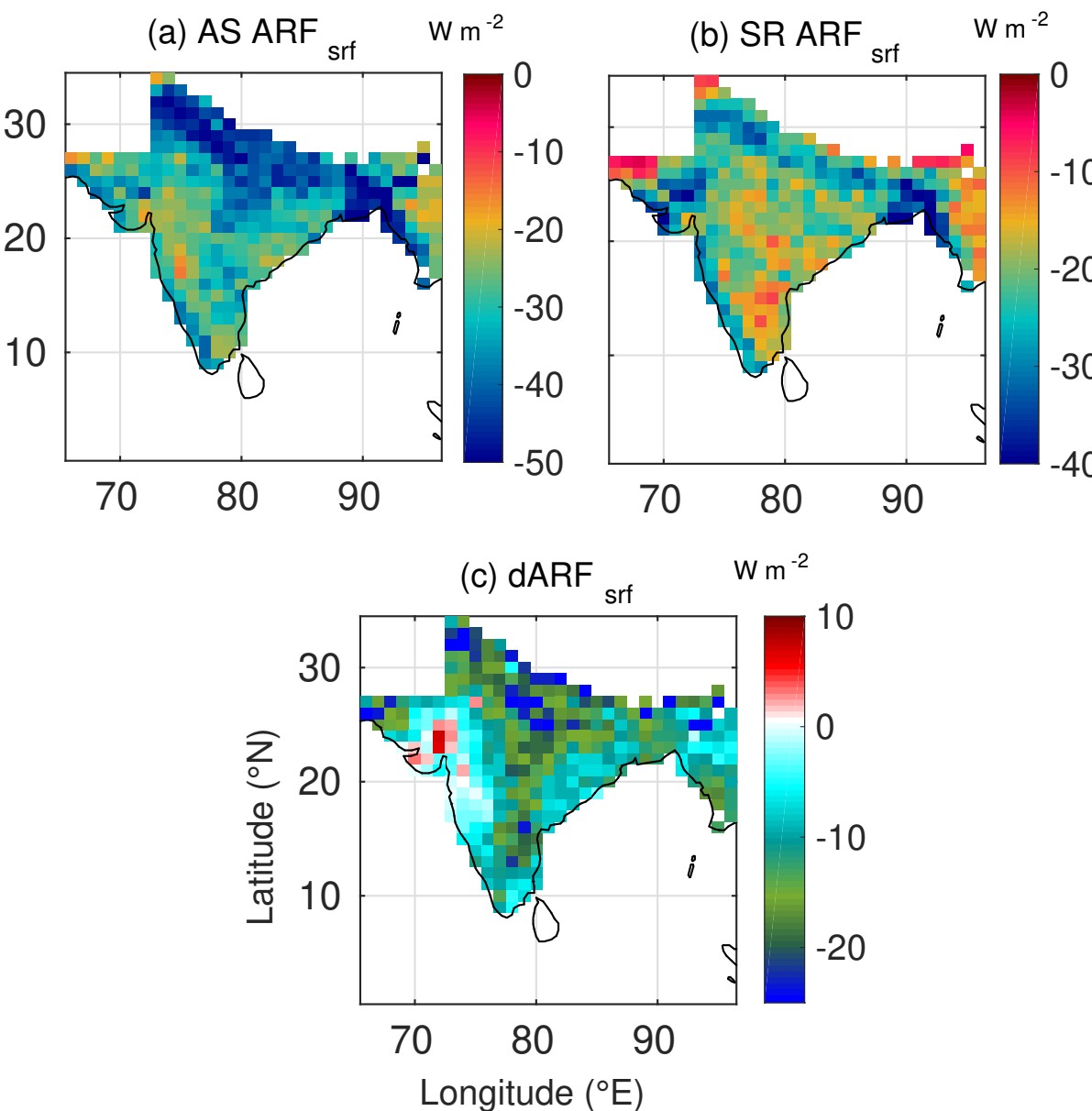

**Figure 6.** Spatial Variation of (a) AS ARF_srf (radiative forcing at surface estimated using assimilated AOD and SSA), (b) SR ARF_srf (radiative forcing at surface estimated using satellite-retrieved AOD and SSA), and (c) the difference between both surface forcing estimates (dARF_srf) for May-2009, over the Indian region

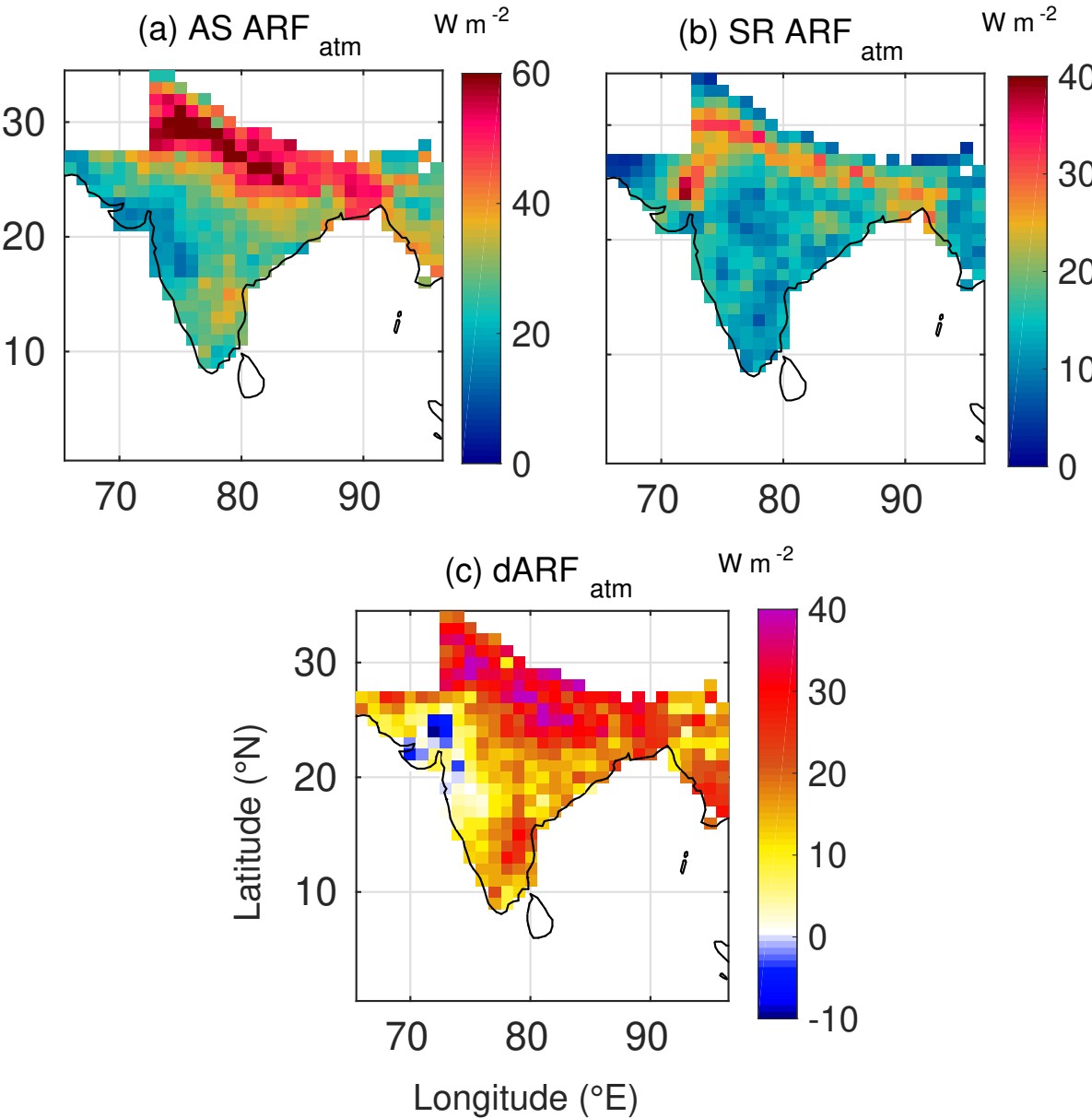

**Figure 7.** Spatial Variation of (a) AS ARF$_{atm}$ (atmospheric forcing estimated using assimilated AOD and SSA), (b) SR ARF$_{atm}$ (atmospheric forcing estimated using satellite-retrieved AOD and SSA), and (c) the difference between both atmospheric forcing estimates (dARF$_{atm}$) for May-2009, over the Indian region

## 4.2 Comparison with CERES measurements

The above described spatial distribution of aerosol direct forcing, and the large season-dependent differences in between the estimates made using assimilated and satellite retrieved datasets calls for a quantitative verification based on independent measurements, which would delineate the datasets that has better accuracy over the study domain. This exercise would also qualify
the superior datasets as inputs to climate models for impact assessment. With a view to accomplishing this, we have compared the TOA fluxes, estimated using the assimilated and satellite-based datasets with spatio-temporally collocated measurements by CERES (Clouds and Earth's Radiant Energy System) onboard AQUA satellite for clear-sky conditions. CERES is a scanning broadband radiometer which measures the upwelling radiances at TOA over three spectral regimes: the shortwave (0.3-5 $\mu$m), the infrared window (8-12 $\mu$m) and the total (0.3 to 200 $\mu$m). These measured radiances are then converted into the
radiative fluxes using the scene-dependent empirical angular distribution models (Loeb et al., 2003), which are then re-gridded to $1° \times 1°$ grid. In the present study, we have used the re-gridded, monthly averaged instantaneous flux measurements at TOA provided by CERES-SSF (Single Scan Footprint) product for clear-sky conditions. However, it is to be noted that the RMS uncertainty ($1\sigma$) corresponding to this monthly averaging of instantaneous shortwave flux measurements is around 9 W m$^{-2}$ (over land). In addition, the monthly CERES SW flux measurements also suffer from the uncertainties arising from those in
calibration of CERES instrument (1 W m$^{-2}$) and radiance to flux conversion process (1 W m$^{-2}$) (Su et al., 2015).

As the equatorial crossing time (ECT) of AQUA (local solar time for ascending orbit) varies slightly about its mean value of 13 : 30 (PRICE, 1991; Ignatov et al., 2004), we have estimated the TOA fluxes using the assimilated and satellite retrieved datasets at time T and $T \pm \sigma_T$ , where T is the mean equatorial crossing time and $\sigma_T$ is the standard deviation in ECTs. The rest of the inputs to SBADRT remained the same for the estimates, as those given in section 3. The TOA fluxes thus
estimated employing assimilated and satellite-based aerosol products are time averaged (over T and $T \pm \sigma_T$ ) for each month and then compared with monthly mean, CERES-measured fluxes. The above estimated, time averaged fluxes corresponding to assimilated and satellite aerosol products are henceforth respectively denoted as AS RAD$_{\text{TOA}}$ and SR RAD$_{\text{TOA}}$ and respective CERES flux measurements are referred as CERES$_{\text{TOA}}$. With a view to quantifying the comparisons, we have computed the deviations of AS RAD$_{\text{TOA}}$ and SR RAD$_{\text{TOA}}$ from corresponding CERES measurements, for each month during the period of 5
25 years (2009-2013)(excluding the monsoon months of JJAS, owing to extensive cloud cover).

$$\delta_{\text{AS}} = \text{AS RAD}_{\text{TOA}} - \text{CERES}_{\text{TOA}} \tag{7}$$

$$\delta_{\text{SR}} = \text{SR RAD}_{\text{TOA}} - \text{CERES}_{\text{TOA}} \tag{8}$$

In the above equation 7 (8), $\delta_{\text{AS}}$ ($\delta_{\text{SR}}$) denotes deviations in the absolute magnitudes of TOA fluxes estimated employing assimilated (satellite retrieved) datasets from the corresponding CERES-measured fluxes. Accordingly, positive values of imply
higher magnitudes of the TOA fluxes estimated using the assimilated(satellite) datasets vis-a-vis CERES measurements. The monthly time series of the RMS (Root Mean Square) of flux deviations ($\delta_{\text{AS}}$, $\delta_{\text{SR}}$) are presented in Figure 8 in the form of

box-whiskers plots, for the entire study period (referred as Annual) as well as for the following three seasons; pre-monsoon (PrM, Mar-Apr-May (MAM) months); winter (Dec-Jan-Feb (DJF) months) and post-monsoon (PoM, Oct-Nov (ON) months) seasons.

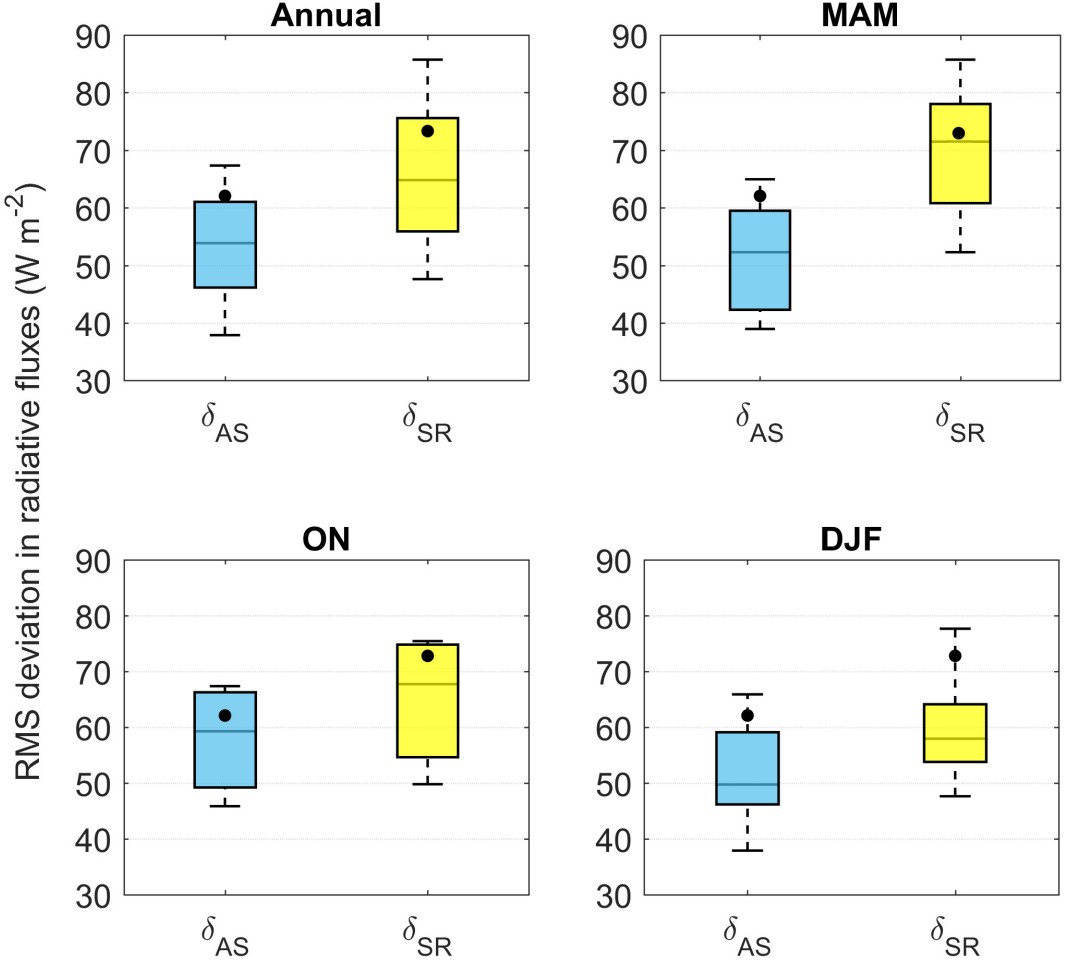

**Figure 8.** Box-whiskers plots for monthly RMS difference between the radiative fluxes at TOA estimated using assimilated products and CERES measurements ($\delta_{AS}$) (blue colored box) and monthly RMS difference between the radiative fluxes at TOA estimated using satellite-based aerosol products and CERES measurements ($\delta_{SR}$) (yellow colored box) for the duration of year 2009 to 2013, over the Indian region. The bottom and top ends of the boxes respectively denote the 25 and 75 percentiles, the median is denoted by black horizontal line within each box, while black circle represents the mean of respective population.

Figure 8 unequivocally shows that median and mean of flux deviations are substantially smaller (at 95% confidence level) for estimates made using the assimilated datasets than when the satellite retrieved datasets are used for all the seasons as well

as for the entire monthly time-series represented by Annual case. However, the difference between the two distributions ($\delta_{AS}$, $\delta_{SR}$) is more conspicuous during the MAM season (figure 8) which is the season of intense aerosol loading over this domain; due to the combined actions of local emissions and long-range transport; being thoroughly mixed and vertically lofted by strong thermal convections. This quantifies the better accuracy of the observationally constrained assimilated, gridded aerosol products over the satellite retrieved aerosol products, over the study domain for inputting in ARF estimation.

However, there are significant differences between TOA fluxes estimated using assimilated products and CERES measurements ($\sim 40$ to $70$ W m$^{-2}$ for annual case, figure 8a). The annual and seasonal mean values of these differences (RMS ($\delta_{AS}$)) and the corresponding CERES$_{TOA}$ measurements are provided the Table 1. It is to be noted that the standard deviation values provided in the Table 1 correspond to annual and seasonal averaging of respective variables over the domain and are representative of the statistical variations.

**Table 1.** Annual and seasonal mean CERES TOA flux measurements and difference between AS RAD$_{TOA}$ and CERES$_{TOA}$ in W m$^{-2}$

|  | Annual | MAM | ON | DJF |
|---|---|---|---|---|
| CERES TOA flux | 133.12±54.06 | 139.29±59.65 | 127.16±42.25 | 126.87±47.68 |
| $\delta_{AS}$ | 61.01±29.78 | 51.51±8.90 | 76.01±43.49 | 58.46±28.53 |

These differences ($\delta_{AS}$) would arise from the uncertainties and biases in AS RAD$_{TOA}$ and CERES$_{TOA}$. However, being constructed from high-quality direct ground-based measurements (Pathak et al., 2019), the systematic bias in the assimilated aerosol properties and hence in AS RAD$_{TOA}$ tend to be very small. Therefore the $\delta_{AS}$ values reported in figure 8 and Table 1 would primarily be due to random errors in AS RAD$_{TOA}$ and CERES$_{TOA}$. In the view of this, we have estimated the uncertainties in AS RAD$_{TOA}$ which primarily originate from those in assimilated AOD and SSA as well as from averaging of AS RAD$_{TOA}$ over the AQUA satellite crossing duration. The uncertainties in the AS RAD$_{TOA}$ due to those in assimilated aerosol products are estimated following the procedure similar to that explained in Appendix A, for the two representative cases of January-2009 and May-2009 and the typical value of RMS uncertainty ($1\sigma$) in AS RAD$_{TOA}$ is around 5.8 W m$^{-2}$. Further, the RMS uncertainty ($1\sigma$) due to temporal averaging of AS RAD$_{TOA}$ over the duration corresponding to expected variation in the satellite crossing time is observed to be 7.6 W m$^{-2}$. The uncertainties in instantaneous TOA flux measurements provided by CERES-SSF (monthly averaged) are already described above.

It can be seen from the above discussion and the Table 1 that the estimated RMS difference between the AS RAD$_{TOA}$ and CERES$_{TOA}$ (i.e. $\delta_{AS}$ which is varying from $\sim 40$ to $70$ W m$^{-2}$ for annual mean case) is substantially contributed by the uncertainties in AS RAD$_{TOA}$ and CERES$_{TOA}$. In addition, the uncertainties associated with MODIS surface reflectance products and the assumed aerosol phase function would also contribute to those in the AS RAD$_{TOA}$ and would reflect in RMS ($\delta_{AS}$). It is to be noted here that although the assimilated aerosol products have demonstrated much better confirmation with independent ground-based direct measurements (Pathak et al., 2019) vis-a-vis satellite-based products, over regions where the ground-based

measurements are less dense or sparse, they would tend to be very close to or nearly same as their satellite counterparts which suffer from substantial uncertainties and biases (Zhang and Reid, 2006; Jethva et al., 2014, 2009) as discussed in Pathak et al. (2019) (the Part -1 paper). As such, to further reduce the differences between the model-estimated (using assimilated products) and CERES measured TOA fluxes, it is required to have denser network of ground-based aerosol measurements. In addition, incorporating spatio-temporally varying aerosol phase function datasets and vertical profiles of aerosol extinction and SSA is expected to reduce the differences between the estimated and measured TOA flux measurements further.

This analysis establishes the improved effectiveness of TOA fluxes estimated using assimilated vis-a-vis satellite products and the similar improvement is implied in the ARF corresponding to assimilated aerosol products than those estimated using their satellite counterparts. In the view of this, we have performed the regional level estimation of the aerosol induced atmospheric heating rate (columnar) using the atmospheric absorption corresponding to assimilated aerosol datasets and compared to those estimated using satellite aerosol products.

## 4.3 Atmospheric heating rate estimation

The atmospheric forcing component of ARF is the amount of energy absorbed by the atmosphere, which heats the atmosphere. The heating rate due to aerosol induced atmospheric absorption is calculated as shown in equation .

$$\frac{\partial T}{\partial t} = \frac{g}{C_p} \frac{\Delta F}{\Delta P} \tag{9}$$

Here, $\frac{\partial T}{\partial t}$ is the atmospheric heating rate (K S$^{-1}$), $g$ is the acceleration due to gravity, $C_p$ is the specific heat of air at constant pressure ($\sim 1005$ J kg$^{-1}$ K$^{-1}$), $\Delta F$ is the aerosol-induced atmospheric absorption and $P$ is the atmospheric pressure. Here, $\Delta P$ is considered to be 300 hPa (pressure varying from 1000 hPa to 700 hPa) implying that atmospheric aerosols are largely concentrated in the first 3 km above the ground which is also supported by observations (Parameswaran et al., 1995; Müller et al., 2001).

The spatial distribution of diurnally averaged atmospheric heating rates estimated using assimilated datasets (HR$_{AS}$) and its difference from similar estimates made using satellite retrieved aerosol datasets (HR$_{AS}$ - HR$_{SR}$) are shown in figure 9, for two representative months, January-2009 and May-2099.

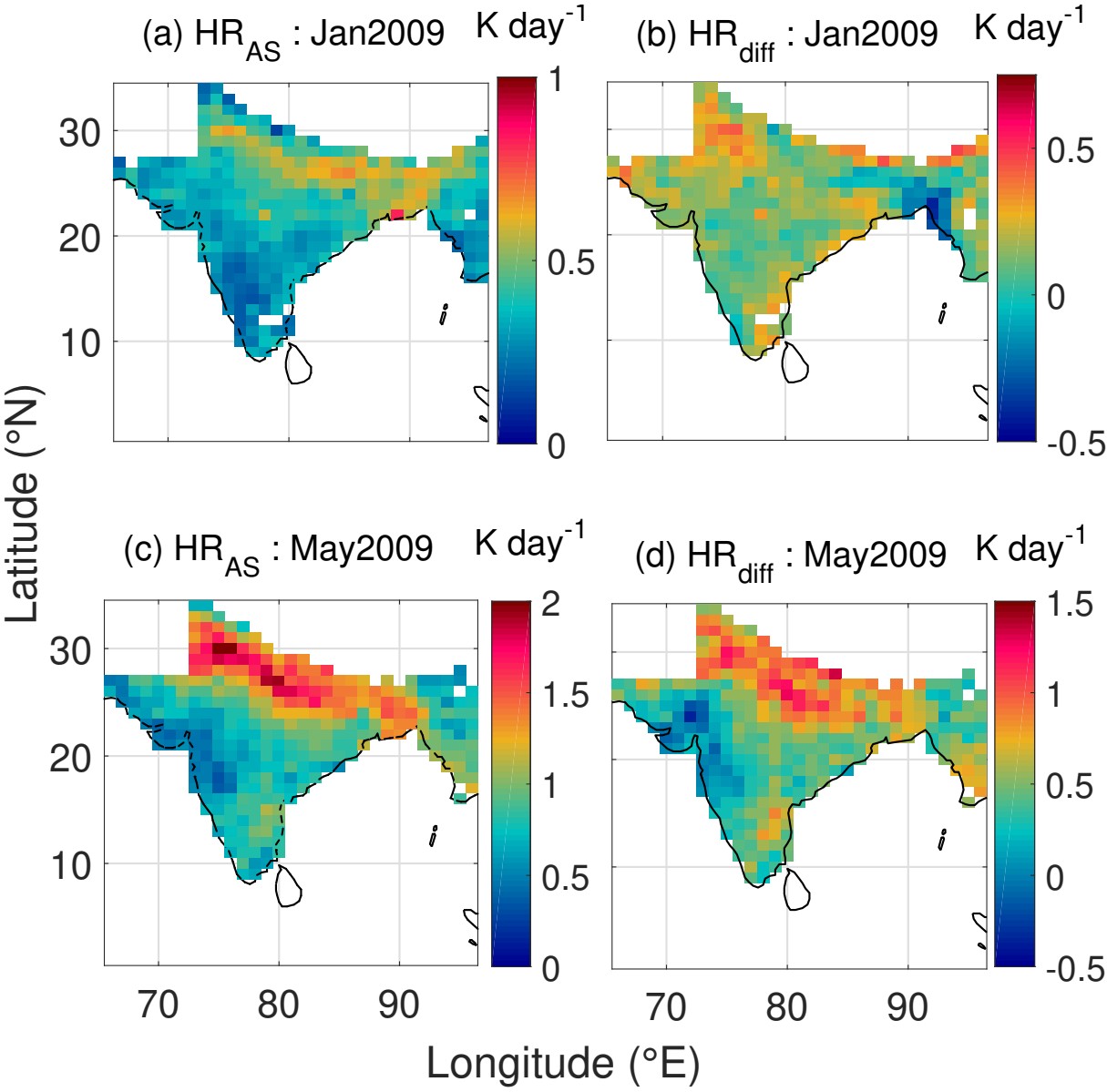

**Figure 9.** Spatial Variation of aerosol-induced atmospheric heating rate (in K day$^{-1}$) estimated using assimilated aerosol products (HR$_{AS}$) for January -2009 and May-2009 (a and c respectively), difference between aerosol-induced atmospheric heating rate corresponding to assimilated and satellite aerosol products (HR$_{diff}$ = HR$_{AS}$ − HR$_{SR}$) for January -2009 and May-2009 (b and d respectively), over the Indian region

The figure reveals consistently higher heating rates ($\sim$ 0.6 to 0.7 K day$^{-1}$ during Jan-2009 and $\sim$ 1.5 to 2.0 K day$^{-1}$ during May) over the Indo-Gangetic plains (IGP) than rest of the sub-regions. As indicated by positive values of HR$_{diff}$, the heating rate estimates using assimilated datasets are consistently higher than those estimated using satellite derived aerosol products.

Further examination of figure 9 reveals that there is significant seasonality in the atmospheric heating rate with pre-monsoonal month, May exhibiting substantially stronger warming than that during the winter month of January, over most of the Indian region. In view of this, we have examined the seasonal variation in the aerosol radiative forcing over the Indian region, in the next section.

5   ## 4.4   Seasonal and sub-regional features

Aerosol types and their properties over Indian region are known to exhibit significant seasonal variation, both at regional and sub-regional scales, primarily due to seasonality in the nature of aerosol sources, advection pathways as well as synoptic and meso-scale meteorology (Jethva et al., 2014; Moorthy et al., 2007a; Babu et al., 2013; Vaishya et al., 2018; Pathak et al., 2019). We examine the signatures of these in direct ARF. For this analysis, we have considered four sub-regions of the spatial domain

10   based on he homogeneity of broad-scale geographical features as detailed in Figure 10 and Table 2.

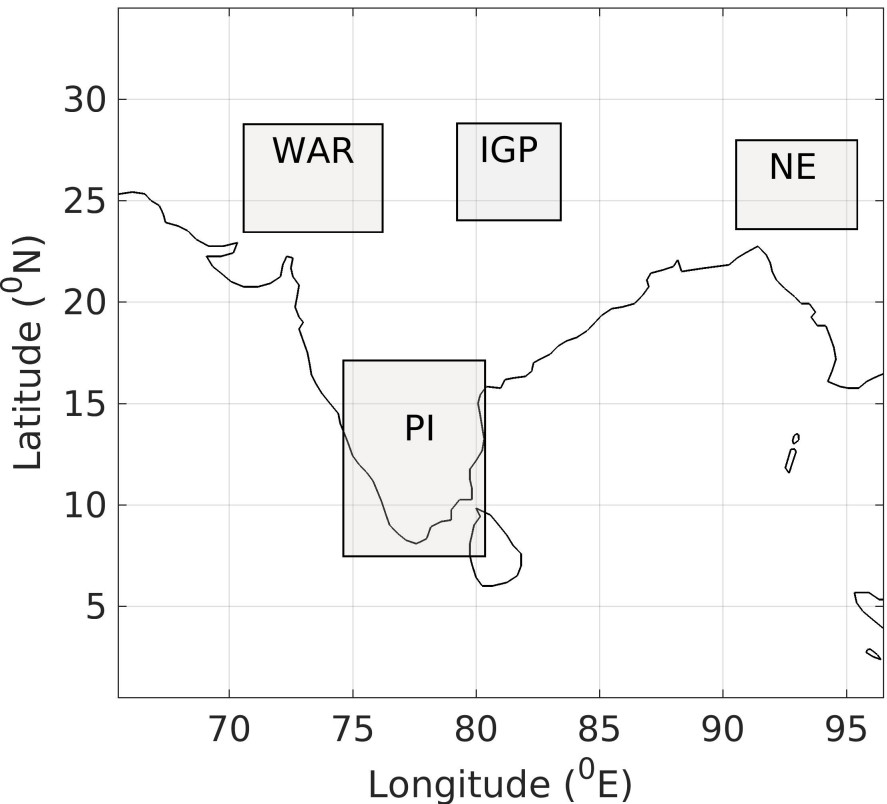

**Figure 10.** Sub-regions of the spatial domain

**Table 2.** Details of sub-regions considered

| Sr no. | Subregion ID | Subregion name | Broad geographical characteristics | Latitudinal boundaries in deg. North | Longitudinal boundaries in deg. East |
|--------|--------------|----------------|------------------------------------|--------------------------------------|--------------------------------------|
| 1 | IGP | Indo-Gangetic plains | Plain plateau | 24.5–28.5 | 78.5–83.5 |
| 2 | NE | North-Eastern India | Mountainous | 23.5–27.5 | 90.5–95.5 |
| 3 | PI | Peninsular India | Coastal and plain | 7.5–17.5 | 74.5–80.5 |
| 4 | WAR | Western Arid Regions | Arid | 23.5–28.5 | 70.5–76.5 |

The climatological seasonal variation of TOA forcing and atmospheric absorption estimated employing assimilated aerosol products and averaged over four sub-regions are presented in Figures 11 and 12 respectively, with the panels a to d respectively representing the IGP, NE, PI and WAR. Please note that the vertical lines over the bars in Figures 11 and 12 indicate the spatio-temporal variation in ARF over the respective sub-region and not the uncertainties in radiative forcing.

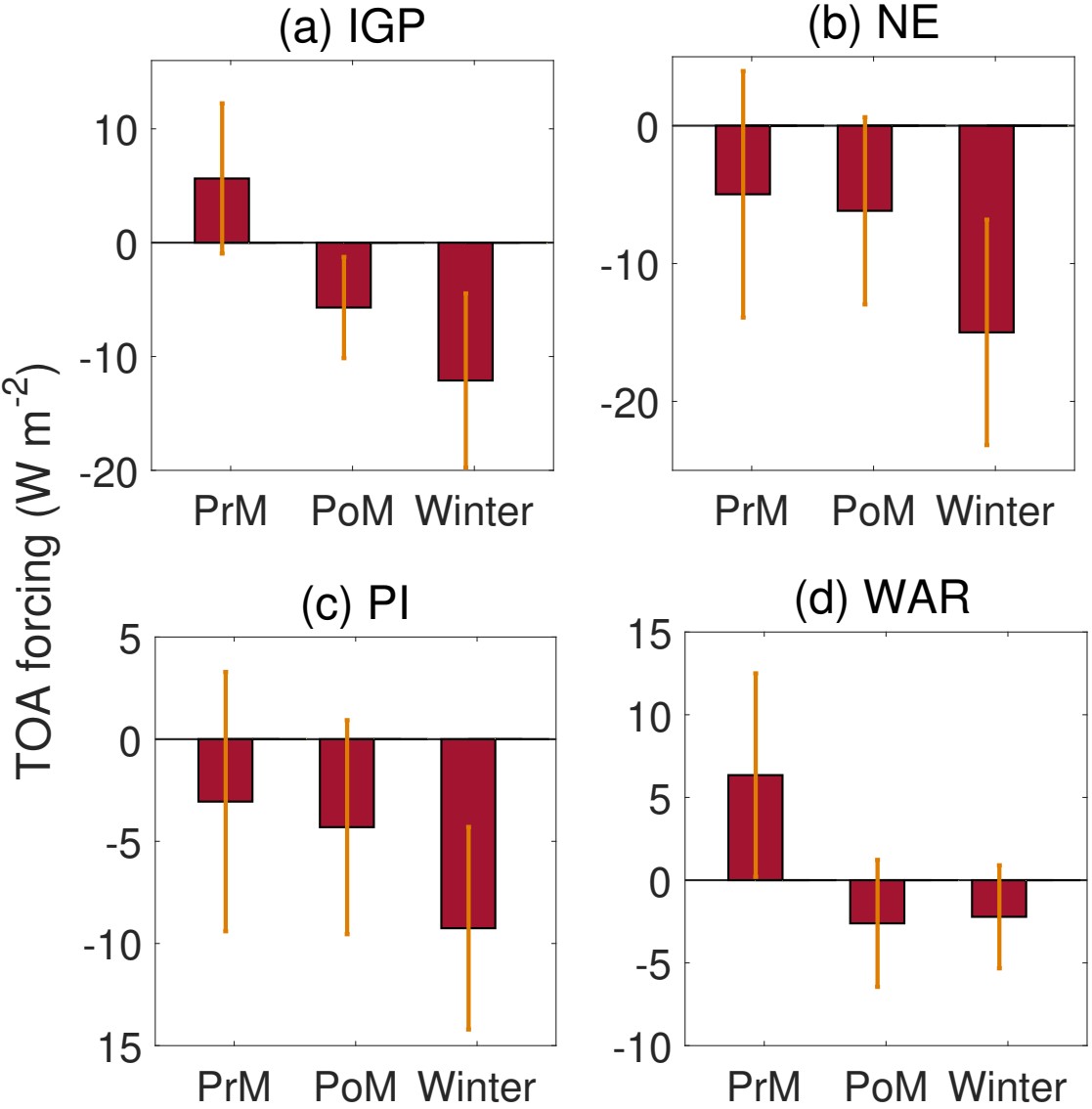

**Figure 11.** The climatological seasonal variation of aerosol radiative forcing at TOA estimated using assimilated aerosol products and averaged over four sub-regions; (a) IGP, (b) NE, (c) PI and (d) WAR

Figure 11 clearly demonstrates that:

1. The TOA forcing is mostly negative over all the sub-regions and most of the seasons, except over the two sub-regions (IGP and WAR), which are highly influenced by advected and locally emitted dust (Banerjee et al., 2019) during pre-monsoon seasons.

2. Seasonally, the magnitude of TOA forcing is highest during winter and least during pre-monsoon over most of the sub-regions, except WAR (Figure 11d) for which the highest magnitude of TOA forcing occurs during pre-monsoon and least during winter, again primarily due to the influence of mineral dust.

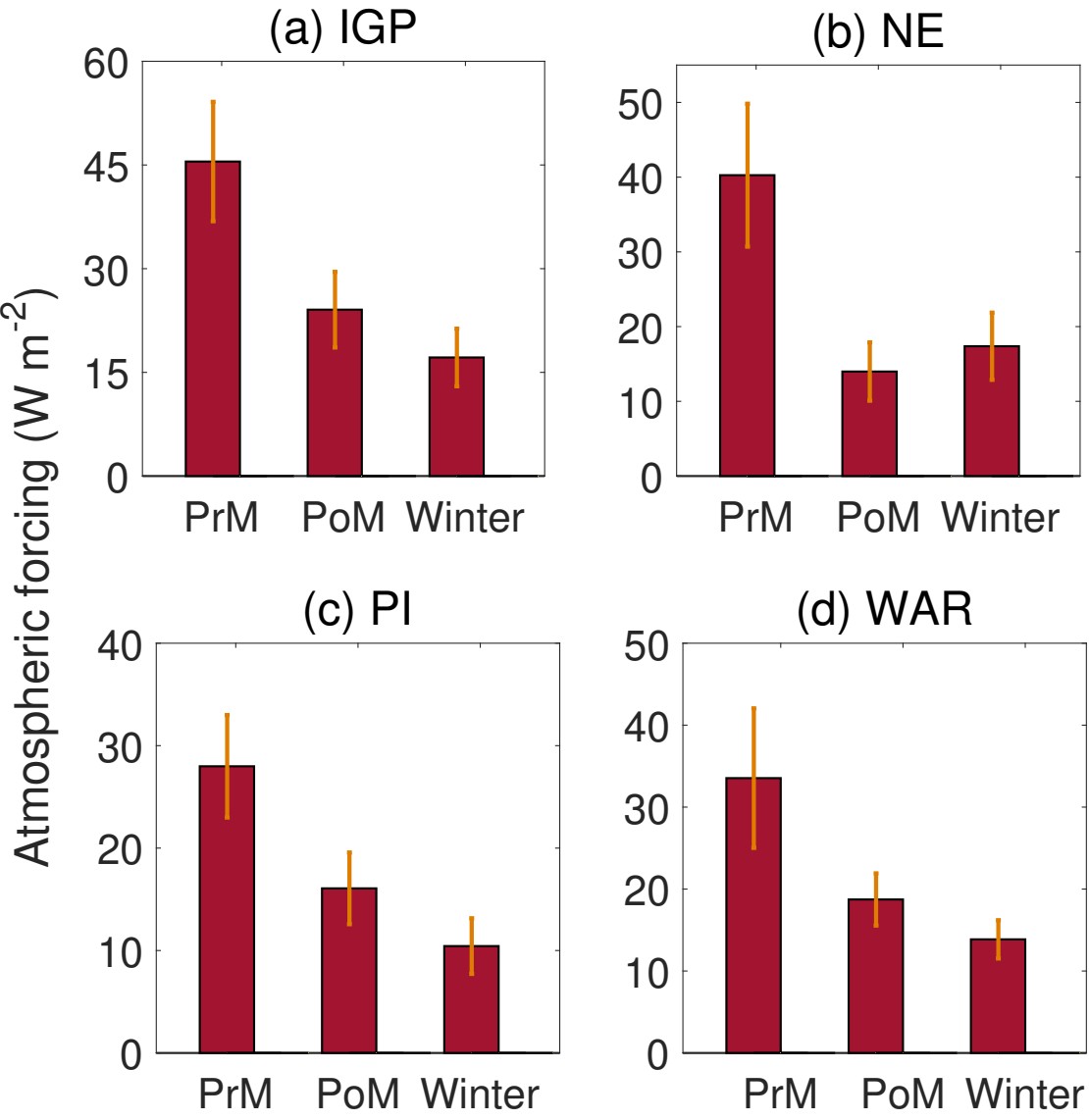

**Figure 12.** The climatological seasonal variation of atmospheric absorption due to aerosols estimated using assimilated aerosol products and averaged over four sub-regions; (a) IGP, (b) NE, (c) PI and (d) WAR

Almost in agreement with the TOA forcing, atmospheric absorption (Figure 12) is maximum during pre-monsoon (when TOA forcing is minimum negative) and least during winter (when TOA forcing is maximum negative) over most of the sub-

regions (except that NE sub-region (Figure 12b) shows least atmospheric absorption during post-monsoon). The highest values of atmospheric forcing though are appearing over IGP (Figure 12a) and least over PI (Figure 12c).

Further, we have estimated the uncertainties in the ARF estimated using assimilated as well as satellite aerosol products and the exercise revealed that the uncertainties in AS ARF are substantially smaller than those in SR ARF which is a direct consequence of smaller uncertainties in assimilated AOD and SSA vis-a-vis respective satellite products (Pathak et al., 2019). The corresponding details are provided in the Appendix A.

## 5    Summary

We have estimated shortwave, clear-sky direct aerosol radiative forcing over the Indian region by incorporating gridded, assimilated, multi-year (2009-2013) datasets for monthly AOD and SSA in SBDART and compared its spatio-temporal features with those in ARF estimated using presently available satellite-retrieved aerosol products. This work is the first of its kind over the Indian region that computes regional ARF estimates that employs assimilated, gridded datasets constrained by highly accurate aerosol measurements performed with a dense network of ground-based observatories spanned across the Indian region. In order to examine the accuracy of these ARF estimates, the monthly instantaneous TOA fluxes estimated using assimilated and satellite retrieved products are compared against the monthly averaged, instantaneous CERES measurements. Finally, we have estimated the aerosol-induced atmospheric warming rates and discussed its spatio-temporal features. The primary findings of the present work are:

1. The TOA fluxes estimated using the assimilated datasets conform better with independent and concurrent space-borne measurements performed by CERES as compared to that shown by TOA fluxes corresponding to satellite retrieved datasets. This establishes the higher accuracy of ARF estimated using assimilated vis-a-vis satellite aerosol products.

2. The diurnally averaged ARF corresponding to assimilated aerosol properties depicts significant spatial and temporal diversity not only in terms of the magnitudes but also in the sign of ARF at TOA. Indo-Gangetic plains, north-eastern parts as well as southern parts of peninsular India exhibit either negative forcing with smaller magnitudes or positive forcing, as compared to rest of the region demonstrating negative TOA forcing of relatively larger magnitudes.

3. The regional distribution of radiative forcing also reveals increased surface cooling and atmospheric absorption over Indo-Gangetic plain and arid regions from north-western India vis-a-vis rest of the region.

4. Similar large-scale spatial features are also shown by ARF estimated using satellite products, however they differ from their assimilated counterparts in terms of magnitude as well as sign of TOA forcing.

5. In consonance with the aerosol induced atmospheric forcing, the atmospheric warming rates exhibit significant spatial heterogeneity ($\sim 0.2$ to $2.0$ K day$^{-1}$), with Indo-Gangetic region demonstrating largest values ($\sim 0.6$ to $2.0$ K day$^{-1}$). In most of the cases, the heating rates corresponding to assimilated products demonstrate substantially increased lower-tropospheric warming vis-a-vis those corresponding to satellite aerosol products.

6. Over most parts of the region, the TOA forcing is negative throughout the year with maximum magnitudes occurring during winter and minimum during pre-monsoon, except Indo-Gangetic plain and western arid regions over which the sign of TOA forcing flips from positive (during pre-monsoon) to negative (during post-monsoon and winter). However, the atmospheric forcing due to aerosols is highest during pre-monsoon and lowest during winter, over almost entire Indian region.

7. The uncertainties in ARF estimated using assimilated aerosol products are substantially lower than those in ARF estimated using satellite products, which is a natural consequence of smaller uncertainties in assimilated vis-a-vis satellite aerosol products.

On the background of these benefits, the present ARF estimates and the corresponding assimilated aerosol products can be potentially applied for improving the accuracy of aerosol climate impact assessment at regional, sub-regional and seasonal scales.

**Appendix A:  Uncertainties in ARF**

One of the prime challenges in the accurate climate impact assessment of aerosols is posed by the uncertainties in the estimation of direct aerosol radiative effect. These uncertainties primarily emanate from those in the gridded datasets for aerosol properties, mainly AOD and SSA. Past studies have shown that small changes in SSA can even change the sign of aerosol radiative forcing at TOA (Haywood and Shine, 1995, 1997; Heintzenberg et al., 1997; Russell et al., 2000; Takemura et al., 2002; Loeb and Su, 2010; Babu et al., 2016). Against this backdrop, it becomes imperative to assess the uncertainties in the radiative forcing estimates presented in the current study.

For estimation of uncertainties in aerosol radiative forcing, we have derived multiple simulations of diurnally averaged ARF (at TOA, surface and within atmosphere) at each grid point over the Indian region, with each of these ARF simulations corresponding to a particular AOD and SSA which are perturbed from the original values within their respective uncertainty limits. The uncertainties in assimilated AOD and SSA are estimated as discussed in Part-1 of the two-part paper (Pathak et al., 2019). The uncertainties in SR AOD (which largely comprises of MODIS AODs) are estimated as $\pm(0.03 + 0.2\tau_M)$ (Sayer et al., 2013) where $\tau_M$ is the corresponding MODIS AOD. The uncertainty in OMI SSA is considered to be $\pm 0.05$ following (Torres et al., 2002) and (Jethva et al., 2014). For a given grid-point, the standard deviation across the multiple simulations of ARF is then considered to be the uncertainty in the radiative forcing estimate. Thus, we have estimated the uncertainties in ARF correspoding to assimilated and satellite-based aerosol products.

The RMS uncertainty in the ARF at TOA, surface and atmospheric absorption estimated using assimilated aerosol products are presnted and compared with those in ARF estimated using satellite derived aerosol products in Table A1 and A2, for the two representative months, January-2009 and May-2009 respectively.

**Table A1.** RMS uncertainty in direct aerosol radiative forcing over the Indian region for January-2009

| | Uncertainty in AS ARF Jan-2009 | Uncertainty in SR ARF Jan-2009 |
|---|---|---|
| TOA forcing | 3.24 | 19.62 |
| Surface forcing | 7.29 | 14.85 |
| Atmospheric Absorption | 9.81 | 12.67 |

**Table A2.** RMS uncertainty in direct aerosol radiative forcing over the Indian region for May-2009

| | Uncertainty in AS ARF May-2009 | Uncertainty in SR ARF May-2009 |
|---|---|---|
| TOA forcing | 2.64 | 13.10 |
| Surface forcing | 6.61 | 10.55 |
| Atmospheric Absorption | 6.24 | 8.49 |

Table A1 and A2 demonstrate that uncertainties in AS ARF are substantially smaller than those in SR ARF, which is the consequence of smaller uncertainties in assimilated aerosol products as compared their satellite counterparts. It can further be seen from Table A1 and A2 that uncertainties in aerosol radiative forcing estimated using assimilated datasets are least for the forcing at TOA as compared to surface forcing and atmospheric absorption. This is in contrast with the ARF corresponding

to satellite products for which the TOA forcing exhibit highest uncertainty vis-a-vis its surface and atmospheric counterparts, which is in line with Chung et al. (2005), in which the global mean TOA forcing (estimated using assimilated product having quite limited signature of ground-based aerosol data) is demonstrated to have higher uncertainties than those in surface and atmospheric forcing. Further inspection of Table A1 and A2 reveals that uncertainties in AS ARF during May-2009 are smaller than those during Jan-2009. This could be primarily because of the assimilated aerosol datasets for May-2009 assimilating

high quality aerosol measurements from more number of stations (19 stations each for AOD and AAOD assimilation) vis-a-vis Jan-2009 (17 and 15 stations for AOD and AAOD assimilation respectively) leading to reduced uncertainties in assimilated AOD and SSA during May-2009 than Jan-2009 and the consequential effect on uncertainties in AS ARF.

Unlike aerosol optical depth (AOD) and single scattering albedo (SSA), there are no extensive ground-based and / or air borne measurements of aerosol size distributions to generate gridded datasets available for estimating the aerosol phase function. As

such, we have used phase functions corresponding to appropriate aerosol models from the Optical Properties of Aerosols and Clouds (OPAC) Hess et al. [1998] in order to estimate ARF using SBDART. The estimated RMS uncertainties are around 2.2

% ($1\sigma$) across the 8 streams of Legendre moments used in the radiative transfer model. While AOD and SSA are the most contributing factors while estimating ARF, phase function plays a relatively less significant role as ARF is the integrated effect over the hemisphere (not angular).

In order to estimate the sensitivity of ARF estimates to the expected variations in the above aerosol phase function, we have simulated the ARF (at Top of Atmosphere (TOA), surface and within atmosphere) for the representative month of January-2009, over the entire Indian region. Each of these simulations were carried out by incorporating Legendre moments of aerosol phase function corresponding to each of the continental aerosol models provided in OPAC along with columnar, assimilated AOD and SSA in SBDART. The uncertainty of ARF w.r.t. that phase function is then estimated as one standard deviation across the multiple ARF simulations. This analysis has revealed that the RMS uncertainty ($1\sigma$) in ARF at TOA, surface and in atmosphere is around 4 %, 0.22 % and 0.05 % respectively. This analysis shows that the present ARF estimates corresponding to assimilated aerosol products are substantially robust w.r.t. expected variations in aerosol phase function; however further improvement in accuracies of the ARF estimation is possible, when realistic size distribution data are generated in future.

*Author contributions.* H

SP carried out estimation of aerosol radiative forcing and further analysis of the data, under the guidance of SKS, KKM and RSN. HSP was also primarily responsible for writing the manuscript which is further reviewed and edited by KKM, SKS and RSN. The valuable inputs regarding aerosols radiative forcing estimation were provided by SKS and KKM.

*Competing interests.* The authors declare that they have no conflict of interest.

*Acknowledgements.* This work is carried out as a part of the project titled "South West Asian Aerosol Monsoon Interactions (SWAAMI) (Grant No:MM/NERC-MoES- 1/2014/002)"funded by the Ministry of Earth Sciences (MoES), New Delhi. We thank Dr. S. Suresh Babu, Dr. M.R. Manoj and all the ARFINET investigators for the continuous efforts and support provided in maintaining the network as well as in collecting and processing the data. One of the authors (SKS) would like to thank SERB-DST for J.C. Bose Fellowship. The Terra, Aqua MODIS Aerosol Optical Depth Monthly, L3, Global, 1 Deg. CMG data sets were acquired from the Level-1 and Atmosphere Archive and Distribution System (LAADS) Distributed Active Archive Center (DAAC), located in the Goddard Space Flight Center in Greenbelt, Maryland *(https://ladsweb.nascom.nasa.gov/)*. MISR Aerosol Optical Depth, Monthly L3 Global data were obtained from the NASA Langley Research Center Atmospheric Science Data Center. We also acknowledge Global Modeling and Merging Office (GMAO) and the GES DISC for distribution of MERRA data. We are thankful to Prof. S. Lakshmivarahan for his valuable guidance regarding data assimilation techniques. We also thank Dr. Hiren Jethva for providing the OMI data.

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
