# Peer review of "Assessment of Regional Aerosol Radiative Effects under SWAAMI Campaign – PART 2: Clear-sky Direct Shortwave Radiative Forcing using Multi-year Assimilated Data"

_Atmospheric Chemistry and Physics, 2020_

## Short Comment (SC1) · 30 May 2020

South West Asian Aerosols Monsoon Interactions (SWAAMI) a co-ordinated field campaign jointly undertaken by the scientists from India and the United Kingdom , have already brought out the Part -1 observations and as a sequel to this , the current paper is Part-2 of SWAAMI campaign results. The contributors have systematically generated a gridded 10X 10 by incorporating the large network of In-situ observations on AOD and SSA and ensured the whole gridded data is seamlessly assimilated well using the 3-D var operations. At domain level, region level and sub region level data integrity is well

represented using the 2009-2013 years data sets. The study estimated direct short-wave aerosol radiative forcing (ARF) over the Indian region using the above gridded data. Further compared the top-of-the-atmosphere (TOA) radiances estimated using the assimilated data with the radiance values measured by Clouds and Earth's Radiant Energy System (CERES) instrument. The regional heterogeneity and responses in terms of ARF, TOA radiances were evaluated using the assimilated gridded data, which otherwise constrains the domain level evaluations.

The study used the aerosol phase function of AOD and SSA were based on OPAC and the Satellite derived AOD are adopted from MODIS; Terra/Aqua and MISR, where as SSA is from OMI on board AQUA satellite, a monthly mean of SSA at 500 nm.

The ARF is computed using the SBDART and the inputs for surface reflectance were adopted from MODIS derived L-3 products, vertical heterogeneity of Ozone and water vapour is taken from OMI and MODIS for the respective periods appropriately. The other atmospheric gases vertical heterogeneity the SBDART based model out puts as available for tropical environment are employed.

The results were evaluated for ARF using the assimilated gridded data based ARF and Satellite data based ARF and the Differences were over the domain and sub region were evaluated.The results were presented using the Winter season data of January 2009 and for Summer data of May 2009 were presented.

The winter (January) assimilated datasets show stronger surface cooling and higher atmospheric warming than those yielded from the satellite retrieved data, except over IGP region (fig 2a and b).It is clearly seen that satellite derived TOA has not shown any warming where as Assimilated data showed at few regions warming as it apparent the assimilated data is pronouncing the TOA to a tune of 2-5 Wm2 . Where as surface forcing in assimilated data relatedly across domain is lower and Satellite only derived surface forcing appear over estimating.(fig 3a, b , and C ). Contrastingly the atmospheric forcing is in assimilated data pronounced in IGP region where in satellite

derived data it is low.(fig;4a-c)

The summer months data of May 2009 for AS ARF TOA is strikingly higher in IGP region and SR ARF TOA relatively low.(fig 5 a and b). Where as for surface forcing in AS ARF srf it as expected the IGP region is cooler than the SR ARF srf. (fig 6a and b). Similarly the Atmospheric ARF in AS ARF atom is significantly higher in IGP and where as it is underestimated in SR ARF atm.(fig 7 a-b).

The key evaluation is by using the collocated regridded Radiance values of CERES satellite TOA as reference since it has three Onboard sensors, covering Short wave, Infra Red and total wave length regions. This data is evaluated with the above (AS; SR ) data products and the whisker plots have clearly shown error bars are well with in permissible limits.

Heating rates were also calculated considering the pressure levels of 1000 hpa to 700 hpa of potentially 3KM altitude.HR were calculated for both winter (jan) and Summer periods (May) for 2009 and clearly projected HR is higher all along IGP region and Western part of india.(fig 9 a-d).

Overall the investigation of SWAAMI Par-2 and data sets employed appear very demonstrative and intriguing. The scientific paper is certainly a quality investigation deserves appearing the ACP.

However very minor suggestion is to provide a X, Y plot from 2009-2013 for five years along with error bars; though whisker plots are given. This will be made year wise clarity. A minor hunch is that the Tropic of Cancer above region is distinctive over all.The extended application of Assimilated data and gridded data may put In use for NCP models as it deserves. Typo issues at : 4.3: line 15 (2009); 4.1 : line 19 (used)

I strongly recommend the paper for appearing in ACP.

---

## Referee Comment (RC1) · Anonymous Referee #1 · 27 Jun 2020

The manuscript is part-2 of their work. In the first part, they have presented the gridded, assimilated datasets of AOD and SSA over the Indian region, which provide spatio-temporally continuous measurements from a dense network of ground-based aerosol observatories and multi-satellite datasets. These resulting improvement inaccuracies of the gridded products in reproducing the Spatio-temporal characteristics of aerosol properties at sub-regional scales over the Indian domain. In this second part, the team has estimated direct shortwave ARF over the region. Also, a comparison of these estimates is made with similar estimates made using the parent satellite data to demon-

strate the effectiveness of the assimilated data. Further, they have compared the TOA radiances estimated using the assimilated data with the radiance values measured by the CERES instrument, and the seasonal contrast in ARF is then presented for various geographically homogeneous sub-regions. Overall this is excellent work with clear objectives and good scientific presentation; however, it needs some attention to improve the presentation. I am listing below some specific suggestions

Title of the manuscript: The manuscript title needs to be clear to understand the readers. At present, it is not clear the region where they have conducted the study. Authors have included the campaign name (SWAAMI), but that also abbreviation and not mention even full in abstract.

Figure cations: I can also recommend to include more details (e.g., study region, abbreviations) in most of the figure captions. There are many abbreviations, and some places short information so hard to read for general readers.

Specific comments follow. Page 3, Line 13-17: I think authors want to say section 4 rather than section 3. Instead, I can suggest removing the name of sections (i.e., section 3.1, section 3.2, and section 3.3) as it is understandable that included in the result and discussion.

Page 4, lines 6-7, which figure?

Page 4, line 7-12, please add references to support

Page 4, line 12, again which figure?

---

## Referee Comment (RC2) · Anonymous Referee #2 · 2 Jul 2020

General Comments:

This manuscript presents spatial variations of seasonal mean shortwave aerosol direct radiative forcing (ARF) at surface, atmosphere and TOA over the Indian region based on satellite observations and statistical assimilation of data from a network of ground-based observatories and multi-satellite products (methodology described in an earlier paper - Part-I). These are also compared with direct estimations of shortwave fluxes from satellite (CERES) observations. The results clearly show the superiority and reliability of ARF estimated using the assimilated data.

[Figure]

The topic of research is highly relevant, the manuscript is well written, the data is reliable and adequate, the analysis and estimates are robust and the results provide significant advancement to the present understanding on the impact (direct radiative effect) of aerosols over the Indian region. I recommend publication of this manuscript in ACP after considering the following suggestions, which are primarily to provide better clarity on the results presented.

Specific Comments:

1. Page-3, Lines 26-30: Clarify on any seasonal variation of the externally mixed continental aerosol model used here (from which the phase function is obtained). Is it seasonally and spatially varying (as the aerosol type undergoes a significant spatio-seasonal variations, which is also stated in Page-4, Lines 6-14). If same phase unction is used in all seasons, what is the sensitivity (typical values, preferably in percentage) of the estimated ARF to any expected variations in the assumed phase function? SSA and AOD are already described in the manuscript.

2. Clarify on the altitude profile of aerosols use in the RT simulations. Discussion of the results and some of the inferences drawn (e.g. Page-11; Lines 13-18) are based on the altitude profile of aerosols. While the statements in Page-11, Lines 13-18 are valid, it is to be seen if they are the result of the altitude variation of aerosol profiles used in the present RT calculations.

3. Statement on the uncertainty in the CERES-derived instantaneous TOA Shortwave radiative fluxes should be included.

4. Clarity on the estimation of diurnal mean ARF may be provided (like the integration of instantaneous ARF from sunrise to sunset or in terms of solar zenith angle, or otherwise). Equations (1-3) are local time dependent at any given location.

5. Figure-8; Pages 15-16: This needs to be clearly understood. Delta_AS represents the difference between instantaneous AS_RAD_TOA and CERES_TOA (Eq.7). Ide-

ally, this difference would be zero as the CERES fluxes as well as the assimilated AS_RAD_TOA are highly reliable (the former is directly estimated from the observed radiances through appropriate ADMs – which is pivotal in the global radiation budget estimates - while the later account for surface albedo and observed aerosol properties). Any biases in either of them would be very small or insignificant. Hence the RMS differences in Delta_AS (having magnitude of 40-60 Wm-2) would arise from the uncertainties (random errors rather than bias) in CERES fluxes and estimated AS_RAD_TOA. In order to understand this properly, please provide the following: (i) mean of CERES_TOA fluxes for different seasons and annual mean, (ii) corresponding mean differences between instantaneous AS_RAD_TOA and CERES_TOA, (iii) typical uncertainties in AS_RAD_TOA and CERES_TOA and (iv) statement on which factors contributed to Delta_AS shown in Fig.8.

6. Figure 2a; Page 7, Lines 21-23: What led to the positive values of TOA ARF over the east Peninsular India? Low SSA? Over Himalayas, it might be because of high surface reflectance. Over NW India, surface reflectance and low SSA might have contributed. State clearly.

7. Page-7, Line-8: Can the month 'May' be treated as representative of summer and pre-monsoon? See the other parts of the manuscript where summer (JJA) and pre-monsoon (March-May) are clearly discriminated (e.g., Line 27, Page-15).

8. Page-4, Lines-7, 12: Add Figure number (Fig.1)

9. Proper usage of brackets while citing reference (e.g., Page-2, Lines 13, 15).

10. Page-7, Line-10: Change "... atmosphere As..." to "... atmosphere. As..."
* * *

---

## Author Comment (AC1) · 30 Aug 2020

**Response to the short comment from Dr. CBS Dutt**

We are thankful towards Dr. CBS Dutt for providing a comprehensive review of our manuscript, for the positive recommendation and valuable suggestions. The specific comments from Dr. CBS Dutt are shown below in red font and our responses to these are given below in black font.

Comment 1. However very minor suggestion is to provide a X, Y plot from 2009-2013 for five years along with error bars; though whisker plots are given. This will be made year wise clarity.

The box-whisker representations of monthly RMS differences between the outgoing radiative fluxes (shortwave) at Top of Atmosphere (TOA) estimated using the assimilated / satellite aerosol datasets and the CERES measurements (CERES-SSF product) (Figure 8, Page No. 16) at annual as well as seasonal level, provide a comprehensive information of the pertinent statistical details; such as mean, median and the $5^{th}$, $25^{th}$, $50^{th}$, $75^{th}$ and $95^{th}$ percentiles. If converted to X-Y plots some of these statistical information would be lost.

The box-whisker representations provided in Figure 8, Page No. 16 have clearly demonstrated that the RMS differences between the TOA fluxes estimated using the assimilated data and CERES measurements are significantly smaller (at 95 % confidence level) than those between TOA fluxes corresponding to satellite data and CERES measurements at annual as well as seasonal scales during the period of 5 years (2009-2013). We agree with the reviewer that the year-wise comparison of the differences between the estimated and measured radiative fluxes would bring more clarity. However, the year-wise separation of this data substantially reduces the number of data points available for the comparison resulting in weakening of statistical significance of the result. Nonetheless, the year-wise comparisons as suggested by the reviewer would be possible in future with the availability of assimilated aerosol products for sufficiently longer time durations. On the background of this, the year-wise X-Y plots are not included.

Comment 2. Typo issues at : 4.3: line 15 (2009); 4.1 : line 19 (used)

Thanks a lot for pointing out these typo issues. We have corrected these on Page No. 18, Line No. 19 and Page No. 7, Line No. 24 in the revised manuscript.

---

## Author Comment (AC2) · 30 Aug 2020

**Responses to the comments from Anonymous Referee-1**

We appreciate and are thankful to the summary observations on the importance of this work and positive recommendation of both the reviewers. There are several comments intended to improve the clarity of the work and its quality. We have addressed all these comments carefully and our point-by-point responses to these are given below. The referee comments are shown in red font and our responses to these are given below in black font.

Comment 1: The manuscript title needs to be clear to understand the readers. At present, it is not clear the region where they have conducted the study. Authors have included the campaign name (SWAAMI), but that also abbreviation and not mention even full in abstract.

Complying with this, we have modified the title of the manuscript as follows.

*'Assessment of Regional Aerosol Radiative Effects under SWAAMI Campaign – PART 2: Clear-sky Direct Shortwave Radiative Forcing using Multi-year Assimilated Data over the Indian Subcontinent'*

The present research has been performed as a part of South West Asian Aerosol Monsoon Interactions (SWAAMI), a joint Indo-UK field campaign, which was focused at understanding the variabilities in aerosol properties and aerosol–monsoon links over the Indian region. The basic information regarding the research campaign, SWAAMI (South West Asian Aerosol Monsoon Interactions) is now briefly mentioned in the abstract of the manuscript (Page No. 1, Line No. 2-5), as indicated by the reviewer.

Comment 2 : Figure cations: I can also recommend to include more details (e.g., study region, abbreviations) in most of the figure captions. There are many abbreviations, and some places short information so hard to read for general readers.

Thanks a lot for this valuable suggestion. We have now modified the captions of Figures 1-9 on the lines suggested by the reviewer. The study region as well as full-forms of necessary abbreviations are now included in the captions of Figures 1-9.

Comment 3: Page 3, Line 13-17: I think authors want to say section 4 rather than section 3. Instead, I can suggest removing the name of sections (i.e., section 3.1, section 3.2, and section 3.3) as it is understandable that included in the result and discussion.

We agree. It should be "section 4.1, section 4.2 and section 4.3" instead of "section 3.1, section 3.2, and section 3.3". However, the names of the sections are removed from Page No. 3, Line No. 13-19, as suggested by the reviewer.

Comment 4 : Page 4, lines 6-7, which figure?

The correct reference to Figure 1 has been included on Page No. 4, Line No. 10.

Comment 5 : Page 4, line 7-12, please add references to support

The reference to Kompalli et al, (2014) has been included in Page No. 4, Line No. 12 in order to support the statement regarding the seasonality in relation between the mixed layer height and vertical distribution of aerosols. They had analysed the temporal variation of Black Carbon aerosol characteristics and its relation with mixed layer height over central India using balloon based

measurements.

Further, the reference to Moorthy et al., (2005, 2007), Jethva et al., (2005) and Niranjan et al., (2007) have been included in Page No. 4, Line No. 14-15 in order to support the statements regarding seasonal variation in aerosol species observed over the Indian region.

Comment 6 : Page 4, line 12, again which figure?

The reference to Figure 1 has been included on Page No. 4, Line No. 16 from the modified manuscript.

**References**

Kompalli, S. K., Babu, S. S., Moorthy, K. K., Manoj, M., Kumar, N. K., Shaeb, K. H. B., and Joshi, A. K.: Aerosol black carbon characteristics over Central India: Temporal variation and its dependence on mixed layer height, Atmospheric research, 147, 27–37, 2014.

Moorthy, K. K., Sunilkumar, S. V., Pillai, P. S., Parameswaran, K., Nair, P. R., Ahmed, Y. N., Ramgopal, K., Narasimhulu, K., Reddy, R. R., Vinoj, V., Satheesh, S. K., Niranjan, K., Rao, B. M., Brahmanandam, P. S., Saha, A., Badarinath, K. V. S., Kiranchand, T. R., and Latha, K. M.: Wintertime spatial characteristics of boundary layer aerosols over peninsular India, Journal of Geophysical Research: Atmospheres, 11, https://doi.org/10.1029/2004JD005520, http://dx.doi.org/10.1029/2004JD005520, d08207, 2005.

Moorthy, K. K., Babu, S. S., Satheesh, S. K., Srinivasan, J., and Dutt, C. B. S.: Dust absorption over the "Great Indian Desert" inferred using ground-based and satellite remote sensing, Journal of Geophysical Research: Atmospheres, 112, https://doi.org/10.1029/2006JD007690, http://dx.doi.org/10.1029/2006JD007690, d09206, 2007.

Niranjan, K., Sreekanth, V., Madhavan, B. L., and Krishna Moorthy, K.: Aerosol physical properties and Radiative forcing at the outflow region from the Indo-Gangetic plains during typical clear and hazy periods of wintertime, Geophysical Research Letters, 34, https://doi.org/10.1029/2007GL031224, https://agupubs.onlinelibrary.wiley.com/doi/abs/10.1029/2007GL031224, 2007.

---

## Author Comment (AC3) · 30 Aug 2020

**Responses to the comments of Anonymous Referee - 2**

We appreciate and are thankful to the summary observations on the importance of this work and positive recommendation of both the reviewers. There are several comments intended to improve the clarity of the work and its quality. We have addressed all these comments carefully and our point-by-point responses to these are given below. The referee comments are shown in red font and our responses to these are given below in black font.

Comment 1: Page-3, Lines 26-30: Clarify on any seasonal variation of the externally mixed continental aerosol model used here (from which the phase function is obtained). Is it seasonally and spatially varying (as the aerosol type undergoes a signifcant spatio- seasonal variations, which is also stated in Page-4, Lines 6-14). If same phase unction is used in all seasons, what is the sensitivity (typical values, preferably in percentage) of the estimated ARF to any expected variations in the assumed phase function? SSA and AOD are already described in the manuscript.

This is an important point. Unlike aerosol optical depth (AOD) and single scattering albedo (SSA), there are no extensive ground-based and / or air borne measurements of aerosol size distributions available to generate gridded datasets for estimating the aerosol phase function. As such, we have used phase functions corresponding to appropriate aerosol models from the Optical Properties of Aerosols and Clouds (OPAC) [Hess et al. 1998] in order to estimate ARF using SBDART. The estimated RMS uncertainties are around 2.2 % (1σ) across the 8 streams of Legendre moments used in the radiative transfer model. While AOD and SSA are the most contributing factors to ARF, phase function plays a relatively less significant role as ARF is the integrated effect over the hemisphere (not angular).

In order to estimate the sensitivity of ARF estimates to the expected variations in the above aerosol phase function, we have simulated the ARF (at Top of Atmosphere (TOA), surface and within atmosphere) for the representative month of January-2009, over the entire Indian region. Each of these simulations were carried out by incorporating Legendre moments of aerosol phase function corresponding to each of the continental aerosol models provided in OPAC along with columnar, assimilated AOD and SSA in SBDART. The uncertainty of ARF w.r.t. that phase function is then estimated as one standard deviation across the multiple ARF simulations. This analysis has revealed that the RMS uncertainty (1σ) in ARF at TOA, surface and in atmosphere is around 4 %, 0.22 % and 0.05 % respectively. This shows that the present ARF estimates corresponding to assimilated aerosol products are substantially robust w.r.t. expected variations in aerosol phase function; however further improvement in accuracies of the ARF estimation is possible, when realistic size distribution data are generated in future.

This discussion has been included as Appendix A, Page No. 26, Line No. 13-16 and Page No. 27, Line No. 1-12 in the revised manuscript.

Comment 2: Clarify on the altitude profile of aerosols used in the RT simulations. Discussion of the results and some of the inferences drawn (e.g. Page-11; Lines 13-18) are based on the altitude profile of aerosols. While the statements in Page-11, Lines 13-18 are valid, it is to be seen if they are the result of the altitude variation of aerosol profiles used in the present RT calculations.

We highly appreciate this comment, which like the previous one, would help to better characterize the ARF and its vertical variation, which is more useful in climate implication studies. However, in the present work, we have used only columnar AOD and SSA and a typical value of aerosol scale height of 1.45 km. This point is included on Page No. 6, Line No. 5-6 in the revised manuscript.

Recent measurements over the Indian region using aircrafts and balloons have shown that aerosol properties do vary with altitude and this variation has seasonal dependency. However, the available data are not adequate to generate a gridded database for the vertical distribution of SSA and AOD representative for the entire region and for different seasons. We are in the process of collating the available data from airborne measurements over the Indian region, carried out during different campaigns since 2007 and would definitely be attempting this once a strong database emerges. The possibility of this improvement is indicated in the revised manuscript on Page No. 11, Line No. 18-23.

Comment 3: Statement on the uncertainty in the CERES-derived instantaneous TOA Shortwave radiative fluxes should be included.

In the present work, we have employed the shortwave (SW) radiative flux (instantaneous) provided by CERES-SSF1Deg (Single Scan Footprint) product. These TOA flux data are further averaged for a given month in order to estimate the monthly averaged, instantaneous flux for that location/grid point. We have observed that the RMS uncertainty ($1\sigma$) corresponding to this monthly averaging of instantaneous shortwave flux measurements is around 9 W m$^{-2}$ (over land). In addition, the monthly CERES SW flux measurements also suffer from the uncertainties arising from those in calibration of CERES instrument (1 W m$^{-2}$) and radiance to flux conversion process (1 W m$^{-2}$) [Su et al., 2015]. These details are now provided in the revised manuscript (Page No. 15, Line No. 12-15).

Comment 4: Clarity on the estimation of diurnal mean ARF may be provided (like the integration of instantaneous ARF from sunrise to sunset or in terms of solar zenith angle, or otherwise). Equations (1-3) are local time dependent at any given location.

We are thankful to the reviewer for suggesting this addition. Accordingly, the following explanation is included in Page No. 6, Line No. 10-14, 16-20 and 22-25 of the revised manuscript.

 The upward and downward shortwave fluxes at the TOA and surface, (in the wavelength range 0.2 to 4 μm) are computed using SBDART for each hour from 6 am (approximate local sunrise time in IST) to 6 pm (approximate local sunset time in IST) for each grid point. The net radiative fluxes are then estimated (considering upward negative and downward positive) for 'with aerosol' and 'without aerosol' conditions and then ARF is estimated as the difference between the net fluxes for the two conditions as has been described in Equations 1-2.

The ARF values estimated in Equations 1-3, which are specific to a solar-zenith angle (or time of the day) for a given location, are further averaged (over the period of 12 hours) and
then halved in order to estimate the diurnally averaged shortwave ARF for the given grid point.

Comment 5: Figure-8; Pages 15-16: This needs to be clearly understood. As represents the difference between instantaneous AS RAD$_{TOA}$ and CERES$_{TOA}$ (Eq.7). Ideally, this difference would be zero as the CERES fluxes as well as the assimilated AS RAD$_{TOA}$ are highly reliable (the former is directly estimated from the observed radiances through appropriate ADMs - which is pivotal in the global radiation budget estimates - while the later account for surface albedo and observed aerosol properties). Any biases in either of them would be very small or insignificant. Hence the RMS differences in $_{AS}$ (having magnitude of 40-60 Wm-2) would arise from the uncertainties (random errors rather than bias) in CERES fluxes and estimated AS RAD$_{TOA}$. In order to understand this properly, please provide the following: (i) mean of CERES$_{TOA}$ fluxes for different seasons and annual mean, (ii) corresponding mean differences between instantaneous AS RAD$_{TOA}$ and CERES$_{TOA}$, (iii) typical uncertainties in AS RAD$_{TOA}$ and CERES$_{TOA}$ and (iv) statement on which factors contributed to $\delta_{AS}$ shown in Fig.8.

Yes, this is very important. As suggested by the reviewer, the annual and seasonal mean of instantaneous CERES measured TOA fluxes averaged over the Indian region are provided in the Table 1. The details of different seasons considered here are as described in Line No. 29-30, Page No. 15 of the manuscript. The corresponding annual and seasonal mean estimates of difference between the TOA fluxes estimated using assimilated products and CERES measurements ($\delta_{AS}$ = AS $RAD_{TOA}$ - $CERES_{TOA}$) are also provided in the Table 1. It is to be noted that the standard deviation values provided in the Table 1 correspond to annual and seasonal averaging of respective variables over the domain and are representative of the statistical variations.

Table 1: Annual and seasonal mean CERES TOA flux measurements (instantaneous) and difference between AS $RAD_{TOA}$ and $CERES_{TOA}$ in W m $^{-2}$

|  | Annual | MAM | ON | DJF |
|---|---|---|---|---|
| CERES TOA flux | 133.12 ± 54.06 | 139.29 ± 59.65 | 127.16 ± 42.25 | 126.87 ± 47.68 |
| $\delta_{AS}$ | 61.01 ± 29.78 | 51.51 ± 8.90 | 76.01± 43.49 | 58.46 ± 28.53 |

Following the suggestion from the reviewer, we have estimated the uncertainties in AS $RAD_{TOA}$ which primarily originate from those in assimilated AOD and SSA as well as from averaging of AS $RAD_{TOA}$ over the satellite crossing duration. The uncertainties in the AS $RAD_{TOA}$ due to those in assimilated aerosol products are estimated following the procedure similar to that is explained in Appendix A, Line no. 19-27, Page no. 25 from the manuscript for the two representative cases of January-2009 and May-2009 and the typical value of RMS uncertainty (1σ) in AS $RAD_{TOA}$ is around 5.8 W m $^{-2}$. Further, the RMS uncertainty (1σ) due to temporal averaging of AS $RAD_{TOA}$ over the duration corresponding to expected variation in the satellite crossing time is observed to be 7.6 W m $^{-2}$. The details about the uncertainties in instantaneous TOA flux measurements by CERES are as provided in the reply to the 3rd comment from the reviewer.

From the Table 1 and the above discussion of uncertainties, it is clear that the estimated RMS difference between the AS $RAD_{TOA}$ and $CERES_{TOA}$ (i.e. RMS ( $\delta_{AS}$) which is varying from 40 to 70 W m $^{2}$ for annual mean case) is substantially contributed by the uncertainties in AS $RAD_{TOA}$ and $CERES_{TOA}$. In addition, the uncertainties in MODIS surface reflectance datasets and the assumed aerosol phase function would also be implied in the AS $RAD_{TOA}$ and would reflect in RMS ($\delta_{AS}$). It is to be noted here that although the assimilated aerosol products have demonstrated much better confirmation with independent ground-based direct measurements [Pathak et al., 2019] vis-a-vis satellite-based products, over regions where the ground-based measurements are less dense or sparse, assimilated aerosol properties would tend to be very close to or nearly same as their satellite counterparts which suffer from substantial uncertainties and biases [Zhang and Reid, 2006, Jethva et al., 2014, 2009] as discussed in Pathak et al., (2019) (the part -1 paper). As such, to further reduce the differences between the model-estimated (using assimilated products) and CERES measured TOA fluxes, it is required to have denser network of ground-based aerosol measurements. In addition, incorporating spatio-temporally varying aerosol phase function datasets and vertical profiles of aerosol extinction and SSA is expected to reduce the differences further. The above points are included on Page No. 17, Line No. 6-27 and Page No. 18, Line No. 1-6 of the revised manuscript.

Comment 6: Figure 2a; Page 7, Lines 21-23: What led to the positive values of TOA ARF over the east Peninsular India? Low SSA? Over Himalayas, it might be because of high surface reflectance. Over NW India, surface reflectance and low SSA might have contributed. State clearly.

The positive TOA forcing over eastern Peninsular India (Figure 2a, 5a from the manuscript) arises primarily due to lower columnar SSA values (0.7 to 0.85) during winter as well as pre-monsoonal months as demonstrated in Figure 1d and 1h, Page no. 5 of the manuscript. These low SSA values indicate the increased presence of Black Carbon (BC) which can be largely associated with large anthropogenic activities (this region has several major harbours, industries and large urban conglomerates such as Chennai (13.08 N, 80.27 E), Vijayawada (16.51 N, 80.65 E), Visakhapatnam (17.68 N, 83.21 E), Bengaluru (12.97 N, 77.59 E), Bhubaneswar (20.30 N, 85.42 E)). We agree with assessment from the reviewer regarding positive TOA forcing over the dust-dominated, arid regions from north-western India and Himalayan foothills. The discussion regarding this is included in Section 4.1, Page No. 11, Line No. 24-29.

Comment 7 Page-7, Line-8: Can the month May be treated as representative of summer and pre-monsoon? See the other parts of the manuscript where summer (JJA) and pre- monsoon (March-May) are clearly discriminated (e.g., Line 27, Page-15).

We are sorry for this oversight. The month May forms part of the pre-monsoon season and is corrected accordingly on Page No. 7, Line No. 13 of the modified manuscript.

Comment 8. Page-4, Lines-7, 12: Add Figure number (Fig.1)

Thanks a lot for pointing this out. Appropriate figure number has been added on Page No. 4, Line No. 10 and 16 from the revised manuscript.

Comment 9. Proper usage of brackets while citing reference (e.g., Page-2, Lines 13, 15).

Thanks; complied with.

Comment 10. Page-7, Line-10: Change "... atmosphere As..." to "... atmosphere. As..."

We are sorry for this typo. The suggested correction has been incorporated on the Page No. 7, Line No. 15 of the revised manuscript.

**References**

Hess, M., Koepke, P., and Schult, I.: Optical properties of aerosols and clouds: The software package OPAC, Bulletin of the American meteorological society, 79, 831–844, 1998.

Jethva, H., Satheesh, S. K., and Srinivasan, J.: Seasonal variability of aerosols over the Indo-Gangetic basin, Journal of Geophysical Research: Atmospheres, 110, https://doi.org/10.1029/2005JD005938, http://dx.doi.org/10.1029/2005JD005938, d21204, 2005.

Jethva, H., Satheesh, S., Srinivasan, J., and Moorthy, K. K.: How good is the assumption about visible surface reflectance in MODIS aerosol retrieval over land? A comparison with aircraft measurements over an urban site in India, IEEE Transactions on Geoscience and Remote Sensing, 47, 1990–1998, 2009.

Jethva, H., Torres, O., and Ahn, C.: Global assessment of OMI aerosol single-scattering albedo using ground-based AERONET inversion, Journal of Geophysical Research: Atmospheres, 119, 9020–9040, https://doi.org/10.1002/2014JD021672, https://agupubs.onlinelibrary.wiley.com/doi/abs/10.1002/2014JD021672, 2014.

Pathak, H. S., Satheesh, S. K., Nanjundiah, R. S., Moorthy, K. K., Lakshmivarahan, S., and Babu, S. N. S.: Assessment of regional aerosol radiative effects under the SWAAMI campaign – Part 1: Quality-enhanced estimation of columnar aerosol extinction and absorption over the Indian subcontinent, Atmospheric Chemistry and Physics, 19, 11 865–11 886, https://doi.org/10.5194/acp-19-11865-2019, https://acp.copernicus.org/articles/19/11865/2019/, 2019

R. McClatchey, R. Fenn, J. Selby, F. Volz, and Jyneza Garing. Optical properties of the atmosphere. Environ. Res. Pap., 32:Papers No. 411, 108 pp, 08 1972.

Su, W., Corbett, J., Eitzen, Z., and Liang, L.: Next-generation angular distribution models for top-of-atmosphere radiative flux calculation from CERES instruments: validation, Atmospheric Measurement Techniques, 8, 3297–3313, https://doi.org/10.5194/amt-8-3297-2015, https://amt.copernicus.org/articles/8/3297/2015/, 2015

Zhang, J. and Reid, J. S.: MODIS aerosol product analysis for data assimilation: Assessment of over-ocean level 2 aerosol optical thickness retrievals, Journal of Geophysical Research: Atmospheres, 111, https://doi.org/10.1029/2005JD006898, http://dx.doi.org/10.1029/2005JD006898, d22207, 2006